# DiffCut: Catalyzing Zero-Shot Semantic Segmentation with Diffusion Features and Recursive Normalized Cut

Paul Couairon[1,2]    Mustafa Shukor[1]
Jean-Emmanuel Haugeard[2]    Matthieu Cord[1,3]    Nicolas Thome[1]
[1]Sorbonne Université, CNRS, ISIR, F-75005 Paris, France    [2]Thales, TSGF, cortAIx Labs, France
[3]Valeo.ai

## Abstract

Foundation models have emerged as powerful tools across various domains including language, vision, and multimodal tasks. While prior works have addressed unsupervised semantic segmentation, they significantly lag behind supervised models. In this paper, we use a diffusion UNet encoder as a foundation vision encoder and introduce DiffCut, an unsupervised zero-shot segmentation method that solely harnesses the output features from the final self-attention block. Through extensive experimentation, we demonstrate that using these diffusion features in a graph based segmentation algorithm, significantly outperforms previous state-of-the-art methods on zero-shot segmentation. Specifically, we leverage a *recursive Normalized Cut* algorithm that regulates the granularity of detected objects and produces well-defined segmentation maps that precisely capture intricate image details. Our work highlights the remarkably accurate semantic knowledge embedded within diffusion UNet encoders that could then serve as foundation vision encoders for downstream tasks. *Project page*: https://diffcut-segmentation.github.io

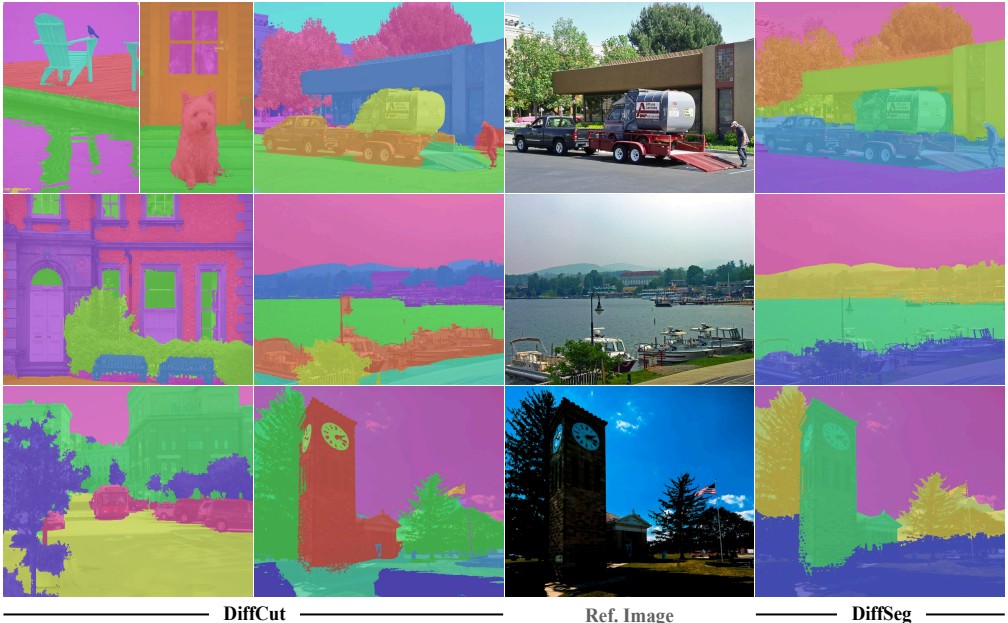

Figure 1: **Unsupervised zero-shot image segmentation.** Our **DiffCut** method exploits features from a diffusion UNet encoder in a graph-based *recursive* partitioning algorithm. Compared to DiffSeg [1], DiffCut provides finely detailed segmentation maps that more closely align with semantic concepts.

38th Conference on Neural Information Processing Systems (NeurIPS 2024).

# 1  Introduction

Foundation models have emerged as powerful tools across various domains, including language [2, 3, 4], vision [5, 6, 7], and multimodal tasks [8, 9, 10, 11, 12, 13]. Pretrained on extensive datasets, these models exhibit unparalleled generalization capabilities, marking a significant departure from training models from scratch to efficiently adapting pretrained foundation models [14, 15, 16, 17]. Utilizing pretrained models is particularly vital for dense visual tasks, alleviating the need for large annotated datasets specific to each domain. While prior works [18, 19, 20, 21, 22] have addressed unsupervised image segmentation, they significantly lag behind supervised models [23, 24, 25, 26]. Recently, SAM [27], proposed a model that can produce fine-grained class-agnostic masks which achieves outstanding zero-shot transfer to any images. Still, it requires a huge annotated segmentation dataset as well as significant training resources. Therefore, in this work, we investigate an alternative direction: unsupervised and zero-shot segmentation under the most constraining conditions, where no segmentation annotations or prior knowledge on the target dataset are available.

Recently, several methods have emerged to address unsupervised object detection by framing it as a graph partitioning problem, utilizing self-supervised ViT features [28, 5]. LOST [29] proposes to localize a unique object in a image by exploiting the inverse degree information to find a seed patch. TokenCut [30] splits the graph in two subsets given a bipartition. FOUND [31] and MaskCut [32] extend these approaches by addressing the single object discovery limitation. While being able to localize multiple objects, the latter methods remain constrained to identify a pre-determined number of objects, making them ill-suited for a task of unsupervised image segmentation which inherently requires to adapt the number of segment to uncover to the visual content.

Conversely, text-to-image diffusion models [33, 34, 35] can produce high-quality visual content from textual descriptions [36, 37, 38], indicating implicit learning of a wide range of visual concepts. Recent works have tried to leverage diverse internal representations of such models for localization or segmentation tasks. Several methods [39, 40, 41, 42, 43] opt to exploit image-text interactions within cross-attention modules but are ultimately constrained by the need for meticulous input prompt design. Concurrently, [44] identifies semantic correspondences between image pixels and spatial locations of low-dimensional feature maps by modulating cross-attention modules. This method proves to be computationally intensive as it requires numerous forward inferences. On the other hand, DiffSeg [1] segment images by iteratively merging self-attention maps which only depict local correlation between patches.

In this work, we introduce DiffCut, a new method for zero-shot image segmentation which solely harnesses the encoder features of a pre-trained diffusion model in a *recursive* graph partitioning algorithm to produce fine-grained segmentation maps. Importantly, our method does not require any label from downstream segmentation datasets and its backbone has not been pre-trained on dense pixel annotations such as SAM [27]. We observe in Fig. 1 that DiffCut produces sharp segments that nicely outline object boundaries. In comparison with the recent state-of-the-art unsupervised zero-shot segmentation method DiffSeg [1], the segments yielded by DiffCut, are better aligned with the semantic visual concepts, *e.g.* DiffCut is able to uncover the urban area as well as the boats in the middle row image. Our main contributions are as follows:

- We leverage the features from the final self-attention block of a diffusion UNet encoder, for the task of unsupervised image segmentation. In this context, we demonstrate that exploiting the inner patch-level alignment yields superior performance compared to merging self-attention maps as done in DiffSeg [1].

- Compared to existing graph based object localization methods *e.g.* TokenCut or MaskCut [30, 32], we push further and take advantage of a *recursive Normalized Cut* algorithm to generate dense segmentation maps. Via a partitioning threshold, the method is able to regulate the granularity of detected objects and consequently adapt the number of segments to the visual content.

- We perform extensive experiments to validate the effectiveness of DiffCut and show that it significantly outperforms state-of-the-art methods for unsupervised segmentation on standard benchmarks, reducing the gap with fully supervised models.

In addition, we exhibit the remarkable semantic coherence emerging in our chosen diffusion features by measuring their patch-level alignment, which surpasses other backbones such as CLIP [8] or

DINOv2 [5]. Our ablation studies further reveal the relevance of these diffusion features as well as the *recursive* partitioning approach which proves to provide robust segmentation performance. Finally, we show that DiffCut can be extended to an open-vocabulary setting with a straightforward process leveraging a *convolutional* CLIP, which even tops most dedicated methods on this task.

## 2 Related Work

**Semantic segmentation.** Semantic segmentation consists in partitioning an image into a set of segments, each corresponding to a specific semantic concept. While supervised semantic segmentation has been widely explored [45, 46, 47, 27], unsupervised and zero-shot transfer segmentation for any images with previously unseen categories remains significantly more challenging and much less investigated. For example, most works in unsupervised segmentation require access to the target data for unsupervised adaption [21, 20, 19, 18]. Therefore, these methods cannot segment images that are not seen during the adaptation. Recently, DiffSeg [1] moved a step forward by proposing an unsupervised and zero-shot approach that can produce quality segmentation maps without any prior knowledge on the underlying visual content.

**Segmentation with Text Supervision.** Recent works have shown that learning accurate segmentation maps is possible with text supervision, overcoming the cost of dense annotations. These works are mostly based on image-text contrastive learning [48, 49, 50, 51], and usually exploit the features of CLIP [52, 53, 54]. MaskCLIP [52] leverages CLIP to get pseudo labels used to train a typical image segmentation model. ReCO [53] uses CLIP for dataset curation and get a reference image embedding for each class that is used to obtain the final segmentation. CLIPpy [48] proposes minimal modifications to CLIP to get dense labels. SegCLIP [54] continues to train CLIP with additional reconstruction and superpixel-based KL loss to enhance localization. TCL [50] learns a region-text alignment to get precise segmentation masks. GroupViT [49] also learns masks from text supervision and is based on a hierarchical grouping mechanism. Similarly, ViewCo [51] proposes a contrastive learning between multiple views/crops of the image and the text.

**Graph-based Object Detection.** Built on top of self-supervised ViT features, various methods frame the problem of object detection as a graph partitioning problem. LOST [29] aims at detecting salient object in an image using the degree of the nodes in the graph and a seed expansion mechanism. Based on *Normalized Cut (NCut)* [55], FOUND [31] proposes to identify all background patches, hence discovering all object patches as a by-product with no need for a prior knowledge of the number of objects or their relative size with respect to the background. TokenCut [30] detects one single salient object in each image with a unique *NCut* bipartition. In an attempt to adapt TokenCut to multi-objects localization, MaskCut [32] first localizes an object and disconnects its corresponding patches to the rest of the graph before repeating the process a pre-determined number of times. As these graph partitioning methods are only able to uncover a fixed number of segments, they are inadequate for a task of image segmentation.

**Segmentation with Diffusion Models.** Diffusion models can produce high-quality visual content given a text prompt, indicating implicit learning of a wide range of visual concepts and the ability of grounding these concepts in images. Therefore their internal representations appear as good candidates for visual localization tasks [56, 57, 58]. ODISE [59] is one of the first training-based approaches to build a fully supervised panoptic image segmentor on top of diffusion features. Several other methods [40, 41, 42] leverage attention modules for localization or segmentation tasks. DiffuMask [42] uses the cross-modal grounding between a text input and an image in cross-attention modules to segment the referred object in a synthetic image. However, DiffuMask can only be applied to a generated image. In a zero-shot setting, [41] harnesses the image-text interaction via cross-attention score maps to complete self-attention maps and segment grounded objects. EmerDiff [44] opts not to exploit image-text interactions in cross-attention modules. Instead, it identifies semantic correspondences between image pixels and spatial locations by modulating the values of a sub-region of feature maps in low-resolution cross-attention layers. These cross-attention based methods eventually prove to be highly computationally intensive as multiple forward inferences are often required. On the other hand, DiffSeg [1] proposes an iterative merging process based on measuring KL divergence among self-attention maps to merge them into valid segmentation masks. However, it appears that self-attention score maps only depict very local correlation between patches.

# 3   DiffCut

**Diffusion Models.**   Diffusion models [60, 61, 62] are generative models that aim to approximate a data distribution $q$ by mapping an input noise $x_T \sim \mathcal{N}(0, I)$ to a clean sample $x_0 \sim q$ through an iterative denoising process. In latent text-to-image (T2I) diffusion models, *e.g.* Stable Diffusion [33], the diffusion process is performed in the latent space of a Variational AutoEncoder [63] for computational efficiency, and encode the textual inputs as feature vectors from pretrained language models. Starting from a noised latent vector $\mathbf{z_t}$ at the timestep $t$, a denoising autoencoder $\epsilon_\theta$ is trained to predict the noise $\epsilon$ that is added to the latent $\mathbf{z}$, conditioned on the text prompt $\mathbf{c}$. The training objective writes:

$$\mathcal{L} = \mathbb{E}_{\mathbf{z} \sim \mathcal{E}(\mathbf{x}), \epsilon \sim \mathcal{N}(0,1), t} \left[ \| \epsilon - \epsilon_\theta(\mathbf{z}_t, t, \tau(\mathbf{c})) \|_2^2 \right] \tag{1}$$

where $t$ is uniformly sampled from the set of timesteps $\{1, \dots, T\}$.

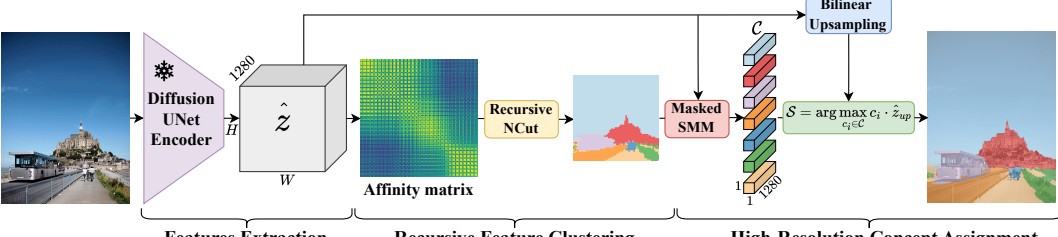

Figure 2: **Overview of DiffCut. 1)** DiffCut takes an image as input and extracts the features of the last self-attention block of a diffusion UNet encoder. **2)** These features are used to construct an affinity matrix that serves in a *recursive normalized cut* algorithm, which outputs a segmentation map at the latent spatial resolution. **3)** A high-resolution segmentation map is produced via a concept assignment mechanism on the features upsampled at the original image size.

## 3.1   Features Extraction

An input image  is encoded into a latent via the VQ-encoder of the latent diffusion model and a small amount of gaussian noise is added to it (not shown in Fig. 2). The obtained latent is passed to the diffusion UNet encoder, from which we only extract the output features, denoted $\hat{z}$, from its last self-attention block.  This choice design has several motivations:

**Attention Limitations.**   In contrast to several methods that harness cross-attention modules for localization or segmentation tasks [39, 40, 41, 42], we deliberately choose not to depend on this mechanism.  The accuracy of segmentation maps generated via attention modules heavily relies on the quality of the textual input which often requires an automatic captioning model combined with a meticulous prompt design to reach competitive performance. Besides being constrained by the maximum number of input tokens, such approach is proved to be inaccurate in the presence of cohyponyms [64] and is prone to neglect subject tokens as the number of objects to detect becomes large [65].  The localization and segmentation capacity with a single forward inference is then constrained by the performance of the captioning model and the attention modules themselves. Exploiting only the intermediate diffusion features alleviate the computational cost of an additional captioning model and do not necessitate multiple forward inferences.

**UNet Encoder Effectiveness.**   Previous works [66, 67, 37] have shown that diffusion features provide precise semantic information shared across objects from different domains. Building on this observation, we hypothesize that the pyramidal architecture of the UNet encoder capture semantically rich image representations that are well-suited for zero-shot vision tasks. To validate this assumption, we exhibit  the *semantic coherence* emerging in the UNet encoder, evidenced by a remarkable patch level alignment in the output features of the final self-attention block. We in fact demonstrate  that these features are sufficient to reach state-of-the-art zero-shot segmentation performance.

**Computational Efficiency.**   By solely exploiting the diffusion UNet encoder, our method offers a substantial computational gain, reducing the model size by 70% (400M *vs* 1.3B parameters). In contrast, DiffSeg extracts every self-attention maps of the UNet which requires a full model inference.

## 3.2 Recursive Feature Clustering

*Normalized Cut* treats image segmentation as a graph partitioning problem [55]. Given a graph $\mathbf{G} = (\mathbf{V}, \mathbf{E})$ where $\mathbf{V}$ and $\mathbf{E}$ are respectively a set of nodes and edges, we construct an affinity matrix $\mathbf{W}$ such that $\mathbf{W}_{ij}$ is the edge between node $v_i$ and $v_j$, and a diagonal degree matrix $\mathbf{D}$, with $d(i) = \sum_j \mathbf{W}_{ij}$. *NCut* minimizes the cost of partitioning the graph into two sub-graphs by solving:

$$(\mathbf{D} - \mathbf{W})x = \lambda \mathbf{D}x \tag{2}$$

to find the eigenvector $x$ corresponding to the second smallest eigenvalue. In the ideal case, the clustering solution only takes two discrete values. Since the solution of Eq. (2) is a continuous relaxation of the initial problem, $x$ contains continuous values and a splitting point has to be determined to partition it. To find the optimal partition, we examine $l$ evenly spaced points in $x$ and select the one resulting in the minimum *NCut* value.

**Graph Affinity.** Based on our observations of the patch-level alignment of diffusion features illustrated in Fig. 4, we assume that the *normalized cut* algorithm will produce sharp segments, each corresponding to a precise semantic concept as distinct objects would manifest as weakly connected components in a patch similarity matrix. Following this intuition, we construct an affinity matrix $\mathbf{W}$, by computing the cosine similarity, normalized between 0 and 1, between each pair of patches. As the *NCut* criterion evaluates both the dissimilarity between different segments and the similarity within each segment, we opt to emphasize inter-segments dissimilarity by raising each element to a positive integer power $\alpha$:

$$\mathbf{W}_{ij} = \left( \frac{\hat{z}_i \hat{z}_j}{\|\hat{z}_i\|_2 \|\hat{z}_j\|_2} \right)^{\alpha} \tag{3}$$

Essentially, this process maintains a relatively high affinity for highly similar patches, while squashing the weights between dissimilar patches towards zero. This mechanism plays the role of a *soft thresholding*, offering a more gradual adjustment compared to setting a threshold to explicitly binarize the affinity matrix as done in [30] and [32].

**Recursive Partitioning.** Classical spectral clustering [68] requires setting a pre-defined number of clusters to partition the graph, which is a significant constraint in the context of zero-shot image segmentation where no prior knowledge on the visual content is available. We therefore adopt a *recursive* graph partitioning [55], which adapts the number of uncovered segments to the visual content via a threshold, denoted $\tau$, on the maximum partitioning cost. This hyperparameter stops the recursive partitioning of a segment when its *NCut* value exceeds it and thereby regulates the granularity of detected objects. We demonstrate that our soft thresholding process detailed above enhances the robustness of the method which delivers competitive performance across a wide range of $\tau$ values. This recursive clustering process is summarized in Supplementary A.

## 3.3 High-Resolution Concept Assignment

Thus far, we have constructed segmentation maps (*e.g.* $32 \times 32$) which are 32 times lower in resolution than the original image (*e.g.* $1024 \times 1024$). The number of segments found in each image depends on the image and the value of the hyperparameter $\tau$. Our next goal is to upscale these low-resolution maps to build accurate pixel-level segmentation maps. We propose a high-resolution segmentation process that can be decomposed into the following steps:

1. **Masked Spatial Marginal Mean.** First, our objective is to extract a set of representations that embeds the semantics of each segment. As shown in [37], reducing the spatial dimension of diffusion features with a *Spatial Marginal Mean* (SMM) effectively retains semantic information and provides accurate image descriptor. In light of this, we naturally propose to collapse the spatial dimension of each segment with a *Masked SMM*. This process yields a collection of semantically rich concept-embeddings, denoted $\mathcal{C}$.

2. **Concept Assignment.** A naive approach to obtain segmentation maps at the original image resolution consists in performing a *nearest-neighbor* upsampling. Despite its straightforwardness, this approach results in a blocky output structure as all pixels within the same feature patch are assigned to the same concept. Alternatively, we opt to first bilinearly upsample our low-resolution feature map $\hat{z}$ to match the original image spatial size and then proceed with the pixel/concept

assignment. Specifically, for each concept $c_i \in \mathcal{C}$, we compute its cosine similarity with the upsampled features $\hat{z}_{up}$. This yields a similarity matrix of size $(H \times W \times K)$ where $K = |\mathcal{C}|$. Then, the assignment process simply consists in taking the argmax across the $K$ channels. The obtained segmentation map $\mathcal{S}$ is eventually refined with a pixel-adaptive refinement module [69].

## 4 Experiments

**Datasets.** Following existing works in image segmentation [21, 53, 52, 18], we use the following datasets for evaluation: **a)** Pascal VOC [70] (20 foreground classes), **b)** Pascal Context [71] (59 foreground classes), **c)** COCO-Object [72] (80 foreground classes), **d)** COCO-Stuff-27 merges the 80 things and 91 stuff categories in COCO-stuff into 27 mid-level categories, **e)** Cityscapes [73] (27 foreground classes) and **f)** ADE20K (150 foreground classes) [74]. An extra background class is considered in Pascal VOC, Pascal Context, and COCO-Object. We ignore their training sets and directly evaluate our method on the original validation sets, at the exception of COCO for which we evaluate on the validation split curated by prior works [21, 19].

**Metrics.** For all datasets, we report the mean intersection over union (mIoU), the most popular evaluation metric for semantic segmentation. Because our method does not provide a semantic label, we use the Hungarian matching algorithm [75] to assign predicted masks to a ground truth mask. For datasets including a background class, we perform a *many-to-one* matching to the background label (Supplementary H). As in [1], we emphasize *unsupervised adaptation* (**UA**), *language dependency* (**LD**), and *auxiliary image* (**AX**). **UA** means that the specific method requires unsupervised training on the target dataset. Methods without the **UA** requirement are considered zero-shot. **LD** means that the method requires text input, such as a descriptive sentence for the image, to facilitate segmentation. **AX** means that the method requires an additional pool of reference images or synthetic images.

**Implementation details.** DiffCut builds on SSD-1B [35], a distilled version of Stable Diffusion XL [34]. The model takes an empty string as input and we set the timestep for denoising to $t = 50$. To ensure a fair comparison when evaluating our method against baselines, we set a unique value for $\tau$ and $\alpha$ across all datasets, equal to 0.5 and 10 respectively. Following previous works, we make use of PAMR [69] to refine our segmentation masks. Our method runs on a single NVIDIA TITAN RTX (24GB) with input images of size $1024 \times 1024$ and can segment an image in one second.

### 4.1 Results on Zero-shot Segmentation

Tab. 1 reports the mIoU score for each baseline across the 6 benchmarks. Note that the numbers shown for COCO-Stuff and Cityscapes are taken from [1]. We complete ReCo [53] and MaskCLIP [52] scores with the results obtained in [50]. Other numbers are taken from [18]. We also note that DiffSeg tunes the sensible merging hyperparameter on a subset of images from the training set from the respective datasets. For a fair comparison, we evaluate the method fixing it to 1, as recommended in the original paper, and refine the obtained masks with PAMR. This baseline is denoted DiffSeg†.

Table 1: **Unsupervised segmentation results.** Best method in **bold**, second is underlined.

| Model | LD | AX | UA | VOC | Context | COCO-Object | COCO-Stuff-27 | Cityscapes | ADE20K |
|---|---|---|---|---|---|---|---|---|---|
| *Extra-Training* | | | | | | | | | |
| IIC [19] | ✗ | ✗ | ✓ | 9.8 | - | - | 6.7 | 6.4 | - |
| MDC [76] | ✗ | ✗ | ✓ | - | - | - | 9.8 | 7.1 | - |
| PiCIE [21] | ✗ | ✗ | ✓ | - | - | - | 13.8 | 12.3 | - |
| PiCIE+H [21] | ✗ | ✗ | ✓ | - | - | - | 14.4 | - | - |
| EAGLE [77] | ✗ | ✗ | ✓ | - | - | - | 27.2 | 22.1 | - |
| U2Seg [78] | ✗ | ✗ | ✓ | - | - | - | 30.2 | - | - |
| STEGO [20] | ✓ | ✗ | ✓ | - | - | - | 28.2 | 21.0 | - |
| ACSeg [18] | ✓ | ✗ | ✓ | 53.9 | - | - | 28.1 | - | - |
| *Training-free* | | | | | | | | | |
| ReCO [53] | ✓ | ✓ | ✗ | 25.1 | 19.9 | 15.7 | 26.3 | 19.3 | 11.2 |
| MaskCLIP [52] | ✓ | ✗ | ✗ | 38.8 | 23.6 | 20.6 | 19.6 | 10.0 | 9.8 |
| MaskCut ($k = 5$) [32] | ✗ | ✗ | ✗ | 53.8 | 43.4 | 30.1 | 41.7 | 18.7 | 35.7 |
| DiffSeg [1] | ✗ | ✗ | ✗ | - | - | - | 43.6 | 21.2 | - |
| DiffSeg† | ✗ | ✗ | ✗ | 49.8 | 48.8 | 23.2 | 44.2 | 16.8 | 37.7 |
| **DiffCut (Ours)** | ✗ | ✗ | ✗ | **65.2** | **56.5** | **34.1** | **49.1** | **30.6** | **44.3** |

With our set of default hyperparameters, DiffCut significantly outperforms all other baselines despite not relying on language dependency, auxiliary images or unsupervised adaptation. On average, our method achieves a gain of +7.3 mIoU over the second best baseline. Notably, DiffCut exceeds MaskCut with an average improvement of +9.4 mIoU. Additionally, it outperforms the previous state-of-the-art method in unsupervised segmentation, DiffSeg, by +5.5 mIoU on COCO-Stuff and +9.4 mIoU on Cityscapes. The superiority of DiffCut over these two methods demonstrates our two key contributions: the high quality of our visual features for semantic segmentation and the flexibility of the recursive NCut algorithm in adjusting the number of segments according to the visual content of each image. The effectiveness of our method is further corroborated by our qualitative results shown in Fig. 1. In comparison to DiffSeg, DiffCut provides finely detailed segmentation maps that more closely align with semantic concepts. Additional examples can be found in Supplementary N.

We note here that, as the granularity of annotations varies across target datasets, our fixed set of hyperparameters can not be in the optimal regime on each of them. Therefore, relaxing the condition on prior knowledge about the target dataset, we report in Supplementary G results of DiffCut where $\tau$ is loosely tuned using a small subset of annotated images from the target training split.

## 4.2 Semantic Coherence in Vision Encoders

As good candidates for a task of unsupervised segmentation are expected to be semantically coherent, we conduct a comparison between different families of foundation models on their internal alignment at the patch-level. Selected models include text-to-image DMs (SSD-1B [35]), text-aligned contrastive models (CLIP [79], SigLIP [80]) and self-supervised models (DINO [28], DINOv2 [5]). At the exception of DINO-ViT-B/16, evaluated models are of roughly similar size, approximately 300M parameters for DINOv2, CLIP-ViT-L/14 and SigLIP-ViT-L/16 and 400M for SSD-1B UNet encoder.

As in [81], we collect patch representations from various vision encoders and store their corresponding target classes using the segmentation labels. Given $\hat{z}_i = \mathcal{E}(\mathbf{x_1})_i \in \mathbb{R}^{D_v}$ and $\hat{z}_j = \mathcal{E}(\mathbf{x_2})_j \in \mathbb{R}^{D_v}$, the patch representations of images $\mathbf{x_1}$ and $\mathbf{x_2}$ at respectively index $i$ and $j$, we compute their cosine similarity and use this score as a binary classifier to predict if the two patches belong to the same class. Given $l(\mathbf{x_1})_i$ and $l(\mathbf{x_2})_j$, the labels associated to the patches, if $l(\mathbf{x_1})_{i,j} = l(\mathbf{x_2})_{p,q}$, the target value for binary classification is 1, else 0. We present in Fig. 3 the ROC curve and AUC score for our candidate models. We observe that SSD-1B UNet encoder [35] demonstrates a greater patch-level alignment than any other candidate model

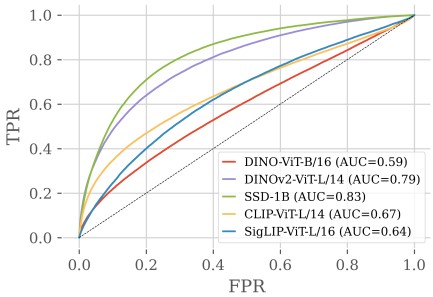

Figure 3: **ROC curves revealing the semantic coherence of vision encoders.**

with an AUC score of 0.83, even surpassing DINOv2 [5]. We further exhibit the outstanding alignment between patch representations associated to semantically similar concepts with qualitative results in Fig. 4. We provide additional qualitative examples patch-level alignment in Supplementary M.

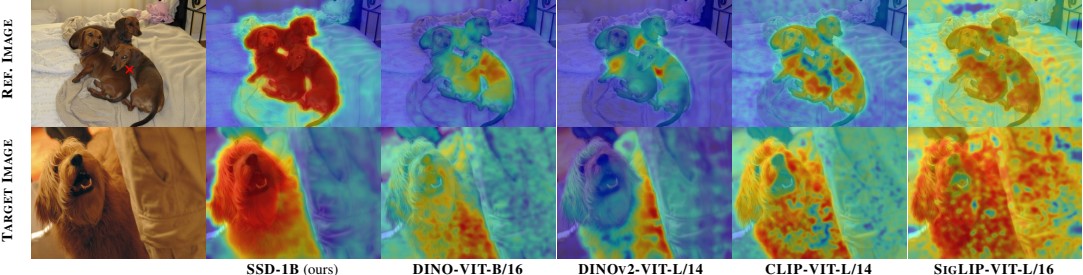

Figure 4: **Qualitative results on the semantic coherence of various vision encoders.** We select a patch (red marker) associated to the dog in **REF. IMAGE**. Top row shows the cosine similarity heatmap between the selected patch and all patches produced by vision encoders for **REF. IMAGE**. Bottom row shows the heatmap between the selected patch in **REF. IMAGE** and all patches produced by vision encoders for **TARGET. IMAGE**.

A potential rationale for this observation lies in the superior semantic information retention of a diffusion model compared to alternative backbones, attributed to its inherent capacity to set a structural image layout, internally acquired during the training phase. These results provides insight into the strong clustering results presented in previous section, as improved semantic coherence suggests that patches belonging to the same object are more effectively clustered.

## 4.3 Ablation study

In this section, we perform ablation studies to validate the individual choices in the design of DiffCut.

**DiffCut *vs* DiffSeg.** DiffSeg proposes their own clustering algorithm based on a self-attention map merging process. As the original implementation uses a different diffusion backbone as ours, we validate the benefit of our method by swapping the original SDv1.4 with our stronger SSD-1B. For a fair comparison between methods, we use the default set of hyperparameters recommended in [1] and set the default merging threshold of DiffSeg to $0.5$ for all datasets. Tab. 2 clearly validates the superiority of using rich semantic features in a *recursive* graph partitioning algorithm over the self-attention merging mechanism of DiffSeg. Qualitative results shown in Fig. 1 further display the edge of DiffCut in uncovering semantic clusters. Shown results do not make use of the mask refinement module, explaining the gap with Tab. 1.

Table 2: **Ablation Study.** The *recursive* partitioning of DiffCut yields superior results to both the self-attention merging process of DiffSeg and Automated Spectral Clustering.

| Model | VOC | Context | COCO-Object | COCO-Stuff-27 | Cityscapes | ADE20K |
|---|---|---|---|---|---|---|
| **DiffSeg** | 48.2 | 41.2 | 31.7 | 35.4 | 22.3 | 39.9 |
| **AutoSC** | 61.5 | 53.3 | 29.8 | **46.9** | 25.3 | 38.9 |
| **DiffCut (w/o PAMR)** | **62.0** | **54.1** | **32.0** | 46.1 | **28.4** | **42.4** |

**Recursive Normalized Cut *vs* Automated Spectral Clustering.** In DiffCut, the hyperparameter $\tau$ corresponds to the maximum graph partitioning cost allowed. In contrast, classical spectral clustering requires to explicitly set the number of segments to be found in the graph. To validate the benefit of the recursive approach over spectral clustering, we introduce a simple yet effective baseline dubbed AutoSC. [82] proposes a heuristic that estimates the number of connected components in a graph with the largest *relative-eigen-gap* between its Laplacian eigen-values. The larger the gap, the more confident the heuristic. In our context, the index of the eigen-value that maximizes this quantity can be interpreted as the number of clusters in an image. Thus, we define a set of exponents $\{1, 5, 10, 15\}$ and determine the value $\alpha$ in this set such that its element-wise exponentiation of matrix $A$ yields the largest Laplacian *relative-eigen-gap*. Then, we use the index of the eigen-value maximizing the gap as the number of clusters in a $k$-way spectral clustering performed with the algorithm proposed in [83]. As shown in Tab. 2, DiffCut consistently outperforms AutoSC on all datasets, with a gain up to $+3.5$ on ADE20K, at the exception of COCO-Stuff where the latter yields slightly better results. Noting that AutoSC is already a state-of-the-art baseline on most benchmarks, this experiment confirms the relevance of the *recursive Normalized Cut* to uncover arbitrary numbers of segments.

## 4.4 Model Analysis

**Hyperparameters Impact.** In this section, we assess the impact of hyperparameters $\tau$ and $\alpha$ over the segmentation performance. We report in Fig. 5 the mIoU for various $\alpha$ values, with respect to partitioning threshold values $\tau$ ranging from 0.3 to 0.97 on Cityscapes validation set. As $\alpha$ increases, we observe a dual effect. First, since a greater $\alpha$ value shrinks the affinity matrix components towards 0, the partitioning cost corresponding to the *NCut* value decreases, explaining the shift of the optimal threshold between the different curves. Second, as $\alpha$ increases, the range of $\tau$ values for which the method yields competitive performance widens, contributing to the overall robustness of the method. For example, DiffCut outperforms our

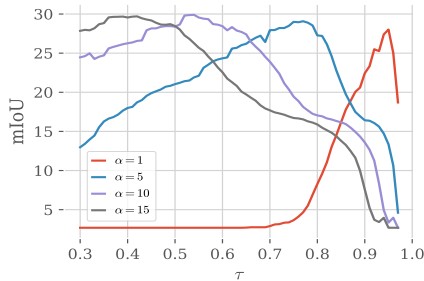

Figure 5: **Sensitivity of DiffCut.** As $\alpha$ increases, DiffCut shows competitive results for a broad range of $\tau$ values.

own competitive baseline AutoSC for any $\tau$ between 0.35 and 0.67 when $\alpha = 10$, whereas it only surpasses it between 0.92 and 0.96 when $\alpha = 1$. Qualitatively, we observe in Fig. 6 that as $\tau$ increases, the method uncovers finer segments in images.

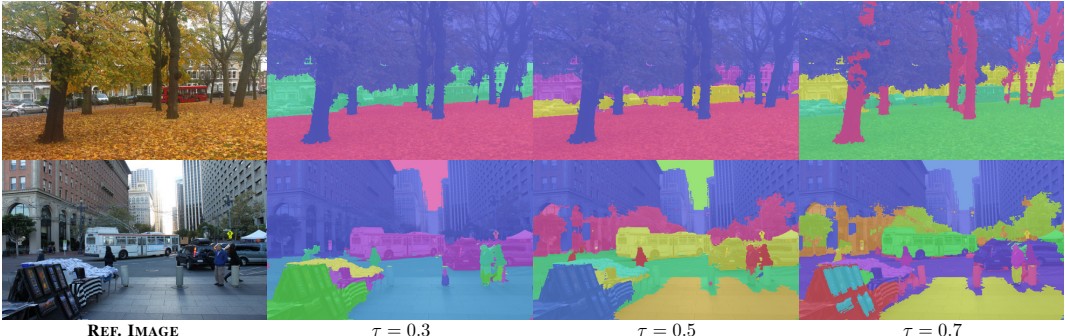

| REF. IMAGE | $\tau = 0.3$ | $\tau = 0.5$ | $\tau = 0.7$ |

Figure 6: **Effect of $\tau$.** As $\tau$ corresponds to the maximum *Ncut* value, a larger threshold loosens the constraint on the partitioning algorithm and allows it to perform more recursive steps to uncover finer objects. It can be interpreted as the level of granularity of detected objects.

**Diffusion Features.** Our chosen diffusion backbone uses a UNet-based architecture which consists of an encoder $\mathcal{E}$, bottleneck $\mathcal{B}$ and a decoder $\mathcal{D}$. The hierarchical features of the encoder, with spatial resolution of $128 \times 128$, $64 \times 64$ and $32 \times 32$ respectively, are injected into the decoder $\mathcal{D}$ via skip connections. Considering the final self-attention modules at resolution $64 \times 64$ and $32 \times 32$ in the encoder and decoder, we demonstrate in Tab. 3, that the encoder's features extracted at the lowest spatial resolution retain the most semantic information and are sufficient to reach optimal performance. In addition, combining different hierarchical features does not lead to any improvements and adds to the computational burden.

Table 3: **Features Contribution**. Hierarchical features in $\mathcal{E}_{32}$ provide optimal performance (Pascal VOC validation set).

| $\mathcal{E}$ | | $\mathcal{D}$ | | VOC Test |
|---|---|---|---|---|
| 32 | 64 | 32 | 64 | **mIoU** |
| ✓ | - | - | - | **62.0** |
| ✓ | ✓ | - | - | 61.6 |
| ✓ | ✓ | ✓ | ✓ | 60.9 |

**Open-Vocabulary Extension.** To extend DiffCut to an open-vocabulary setting, we propose in Tab. 4, a straightforward approach that assigns a semantic label to each segmentation mask. After mask proposals are generated, an image is processed by a frozen *convolutional* CLIP visual encoder, which produces visual representations aligned with text in a shared embedding space via a projection layer. The embedding of each predicted segment is obtained by mask-pooling CLIP visual features, allowing a classification against category text embeddings through contrastive matching. Specifically, let $\mathbf{e}$ represent the embedding of a segment, and let $\{t_i\}_{i=1}^{N}$ denote the text embeddings of category names generated by the pretrained text encoder, the predicted class for this segment is determined as

Table 4: **Open-Vocabulary Segmentation.** A straightforward open-vocabulary extension with a CNN-based CLIP yields competitive performance.

| Model | LD | VOC | Context | COCO-Object |
|---|---|---|---|---|
| *Extra-Training* | | | | |
| ViL-Seg [84] | ✓ | 37.3 | 18.9 | - |
| TCL [50] | ✓ | 55.0 | **30.4** | 31.6 |
| CLIPpy [48] | ✓ | 52.2 | - | 32.0 |
| GroupVIT [49] | ✓ | 52.3 | 22.4 | 24.3 |
| ViewCo [51] | ✓ | 52.4 | 23.0 | 23.5 |
| SegCLIP [54] | ✓ | 52.6 | 24.7 | 26.5 |
| OVSegmentor [85] | ✓ | 53.8 | 20.4 | 25.1 |
| *Training-free* | | | | |
| ReCO [53] | ✓ | 25.1 | 19.9 | 15.7 |
| MaskCLIP [52] | ✓ | 38.8 | 23.6 | 20.6 |
| CLIP-DIY [86] | ✓ | 59.9 | 19.7 | 31.0 |
| FreeSeg-Diff [87] | ✗ | 53.3 | - | 31.0 |
| **DiffCut** | ✗ | **63.0** | 24.6 | **36.0** |

follows: $c = \arg\max_{i \in \{1, \cdots, N\}} softmax([cos(\mathbf{e}, t_1), cos(\mathbf{e}, t_2), \cdots, cos(\mathbf{e}, t_N)])$. This proposed extension reaches competitive performance, even outperforming several baselines dedicated to the task of open-vocabulary zero-shot semantic segmentation.

## 5 Discussion

In this work, we tackle the challenging task of unsupervised zero-shot semantic segmentation by introducing DiffCut, a method that significantly narrows the performance gap with fully supervised

models. DiffCut leverages the diffusion features of a UNet encoder within a recursive graph partitioning algorithm to generate sharp segmentation maps, achieving state-of-the-art results on popular benchmarks. By reusing pre-trained models in a zero-shot manner, our approach can not only reduce computational resources, energy consumption, and human labor but also align with sustainable AI practices. However, because the diffusion backbone is not specifically trained for specialized domains, such as biomedical imaging, the method may struggle with out-of-distribution images. Fine-tuning the diffusion model on domain-specific data could mitigate this challenge. While fully supervised, end-to-end segmentation methods currently offer better efficiency and accuracy, further advancements could close this gap, positioning diffusion-based UNet encoders as foundation models for future vision tasks.

## Acknowledgement

We thank Louis Serrano, Adel Nabli, and Louis Fournier for their insightful discussions and helpful suggestions on the article. This work was granted access to the HPC resources of IDRIS under the allocation 2024-AD011014763 made by GENCI. We acknowledge the financial support provided by ANRT for its funding through the CIFRE grant 2022/0817 and PEPR Sharp (ANR-23-PEIA-0008, ANR, FRANCE 2030).

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

# DiffCut: Catalyzing Zero-Shot Semantic Segmentation with Diffusion Features and Recursive Normalized Cut
## Supplementary Material

## A  Recursive Normalized Cut on Diffusion Features

We summarize in Algorithm 1 the *recursive* clustering process used in DiffCut.

---
**Algorithm 1** Recursive Normalized Cut on Diffusion Features

---
**Input:**  $\mathcal{I}$ an image, $\tau \in ]0, 1[$ a threshold value, $\alpha \in \mathbb{N}^*$ an exponent value.
**Step 1:  Features Extraction.**
- Encode the image $\mathcal{I}$ with the VAE of the diffusion model: $z = \mathcal{E}_{\text{VAE}}(\mathcal{I})$
- Add some gaussian noise to the latent image $z$.
- Pass the noisy latent to the diffusion UNet encoder and extract the output features from its last self-attention block: $\hat{z} = \mathcal{E}_{\text{UNet}}(z)$

**Step 2:  Graph Construction.**
- Compute the pairwise cosine-similarity between patches of $\hat{z}$ to set up a similarity matrix $\boldsymbol{A}$.
- Raise to the power $\alpha$ each element of matrix $\boldsymbol{A}$ to obtain the affinity matrix $\boldsymbol{W}$ (see Eq. (3)).
- Determine the matrix degree $\boldsymbol{D}$.

**Step 3:  NCut Problem Solving.**
- Solve $(\boldsymbol{D} - \boldsymbol{W})x = \lambda \boldsymbol{D}x$ for eigenvector with the second smallest eigenvalue.
- Use the eigenvector with the second smallest eigenvalue to bipartition the graph by finding the splitting point such that the *NCut* value is minimized.

**Step 4:  Recursive Partitioning.**
- Store the current partition and retrieve the matrices $\boldsymbol{W}$ and $\boldsymbol{D}$ associated to each sub-graph.
- Recursively subdivide the partitions **(Step 3)** until the *NCut* value is greater than $\tau$.

**Output:**  $M$ a segmentation map with the spatial resolution of $\hat{z}$.

---

## B  Impact of PAMR

To reveal the effect of the pixel-adaptive refinement module (PAMR) on our method, we compare the segmentation results on all benchmarks both with and without it enabled.

Table 5: **Impact of PAMR on unsupervised segmentation.**

| DiffCut | VOC | Context | COCO-Object | COCO-Stuff | Cityscapes | ADE20K |
|---------|-----|---------|-------------|------------|------------|--------|
| **w/o PAMR** | 62.0 | 54.1 | 32.0 | 46.1 | 28.4 | 42.4 |
| **w/ PAMR** | 65.2 | 56.5 | 34.1 | 49.1 | 30.6 | 44.3 |

In average, PAMR allows to gain +2.5 mIoU on our segmentation benchmarks. Even though this refinement module helps to better outline the contour of objects, our method still reaches state-of-the-art results on unsupervised zero-shot segmentation without it.

## C  Additional Comparison with MaskCut

DiffCut, is capable of providing dense segmentation maps and dynamically adapting the number of detected segments based on the visual content of an image. In contrast, MaskCut [32] can only detect

a fixed number of segments, making it less suitable for image segmentation. This limitation arises from the use of an iterative graph partitioning approach, where graph nodes associated with detected objects are masked. Each segment is treated as a single object and cannot be refined after detection, which severely restricts its ability to identify a large number of objects. To highlight the superiority of our recursive partitioning over MaskCut's iterative process, we present below a comparison between the two methods with $k$ the number of objects to be detected varying in $\{3, 5, 20\}$.

Table 6: **Comparison with MaskCut.** DiffCut *recursive* partitioning algorithm yields superior results than MaskCut iterative partitioning.

| Model | VOC | Context | COCO-Object | COCO-Stuff-27 | Cityscapes | ADE20K |
|---|---|---|---|---|---|---|
| **DiffCut** | **62.0** | **54.1** | **32.0** | **46.1** | **28.4** | **42.4** |
| **MaskCut** ($k = 3$) | 53.7 | 42.3 | 30.9 | 41.8 | 18.0 | 33.7 |
| **MaskCut** ($k = 5$) | 53.8 | 43.4 | 30.1 | 41.7 | 18.7 | 35.7 |
| **MaskCut** ($k = 20$) | 53.8 | 43.5 | 30.0 | 41.5 | 18.0 | 35.6 |

DiffCut significantly outperforms MaskCut, regardless of the chosen value of $k$. DiffCut's improvement shows our two key contributions: the effectiveness of our visual features for semantic segmentation and the ability of the recursive NCut algorithm to dynamically adjust the number of segments based on the visual content of each image.

## D  DiffCut with Alternative Diffusion Backbones

To further display the relevance of diffusion features, we show that DiffCut achieves competitive performance even when using smaller diffusion backbones than SSD-1B. Specifically, we test two alternative models: SD1.4 and SSD-Vega [35] (another distilled version of SDXL). The UNet encoder in SD1.4 has 260M parameters, comprising approximately 30% of the overall UNet, while the UNet encoder in SSD-vega has 240M parameters, making up around 32% of the UNet.

Table 7: **Performance of DiffCut with alternative diffusion backbones.**

| Model | VOC | Context | COCO-Object | COCO-Stuff-27 | Cityscapes | ADE20K |
|---|---|---|---|---|---|---|
| **DiffCut w/ SD1.4** | 57.5 | 52.8 | 30.0 | 45.2 | 24.5 | 36.7 |
| **DiffCut w/ SSD-Vega** | 62.2 | 56.4 | **34.9** | **49.5** | 30.1 | **45.7** |
| **DiffCut w/ SSD-1B** | **65.2** | **56.5** | 34.1 | 49.1 | **30.6** | 44.3 |
| **DiffSeg** | 49.8 | 48.8 | 23.2 | 44.2 | 16.8 | 37.7 |

The results obtained using SD1.4 and SSD-Vega are consistent with those achieved with SSD-1B. While the SD1.4 UNet encoder shows a slight performance drop compared to SSD-1B, DiffCut still significantly outperforms DiffSeg. Notably, with the SSD-Vega UNet encoder, DiffCut delivers performance comparable to SSD-1B, despite having only half the number of parameters.

## E  Mask Upsampling

*Normalized Cut* algorithm does not scale well with the graph size due to the generalized eigenvalue problem to solve, which hinder its use on the native image resolution (*e.g.*, $1024 \times 1024$). Thus, the clustering is applied in the latent space and yields segmentation maps at the latent resolution. To obtain pixel-level segmentation at the original image resolution, we need to

Table 8: **Mask Upsampling.**

| Strategy | VOC Test |
|---|---|
| Concept Assignment | **62.0** |
| Nearest Upsampling | 61.2 |

upscale the low-resolution maps. In Tab. 8, we compare the *nearest-neighbor* upsampling approach versus our concept assignment upsampling and show that our proposed method obtain better results than the naive upsampling of the segmentation masks.

## F  Visual Encoders KMeans Comparison

To evaluate the potential of vision encoders for zero-shot segmentation, we compare their clustering performance with a simple KMeans algorithm. For selected vision encoders, features are extracted

from the last layer. The hyperparameter $K$ is either determined by the ground-truth ($K^*$) for each image, or fixed across the dataset. We compute the mIoU with respect to the ground truth masks using the Hungarian matching algorithm [75]. Tab. 9 shows that the diffusion UNet encoder (**SSD-1B**) significantly outperforms other vision encoders on Pascal VOC (20 classes and no background), COCO-Stuff-27 and Cityscapes. This confirms that diffusion features are well-suited for localizing and segmenting objects. Interestingly, unsupervised models such as DINOv2 are better than CLIP models, suggesting that text-aligned features does not contain accurate localization features.

| Model | $K^*$ | $K=3$ |
|---|---|---|
| **SSD-1B** | **79.5** | **70.8** |
| *Text-aligned* | | |
| CLIP-VIT-B/16 | 68.9 | 59.1 |
| CLIP-VIT-L/14 | 67.1 | 60.7 |
| SigLIP-B/16 | 62.9 | 55.3 |
| SigLIP-L/16 | 62.2 | 54.8 |
| *Unsupervised* | | |
| DINO | 73.1 | 62.8 |
| DINOv2-B/14 | 73.8 | 64.8 |
| DINOv2-L/14 | 73.1 | 64.4 |

(a) VOC

| Model | $K^*$ | $K=6$ |
|---|---|---|
| **SSD-1B** | **36.4** | **37.8** |
| *Text-aligned* | | |
| CLIP-VIT-B/16 | 31.1 | 31.8 |
| CLIP-VIT-L/14 | 26.4 | 26.6 |
| SigLIP-B/16 | 25.0 | 25.1 |
| SigLIP-L/16 | 22.5 | 23.1 |
| *Unsupervised* | | |
| DINO | 33.8 | 34.0 |
| DINOv2-B/14 | 31.5 | 32.2 |
| DINOv2-L/14 | 30.9 | 31.5 |

(b) COCO-Stuff-27

| Model | $K^*$ | $K=6$ |
|---|---|---|
| **SSD-1B** | **21.4** | **21.0** |
| *Text-aligned* | | |
| CLIP-VIT-B/16 | 14.9 | 14.7 |
| CLIP-VIT-L/14 | 14.3 | 14.0 |
| SigLIP-B/16 | 13.7 | 13.6 |
| SigLIP-L/16 | 11.6 | 11.6 |
| *Unsupervised* | | |
| DINO | 18.4 | 17.4 |
| DINOv2-B/14 | 19.5 | 18.9 |
| DINOv2-L/14 | 18.2 | 18.1 |

(c) Cityscapes

Table 9: **KMeans features clustering for various vision encoders.**

## G    Hyperparameter $\tau$ tuning

We show in Sec. 4.3 that the performance of the method is highly robust with respect to the value of the threshold $\tau$. However, as the granularity of annotations varies across target datasets, the value of this threshold, fixed in our experiments, can not be in the optimal regime on each benchmark. Therefore, we relax the condition on the absence of prior knowledge about the target dataset and report in Tab. 10 results of DiffCut where $\tau$ is loosely tuned using a small subset of annotated images (200) from the target training split. Specifically, we estimate an adequate value for $\tau$ with a grid-search in the set $\{0.35, 0.55, 0.75\}$ for COCO-Object, COCO-Stuff and Cityscapes.

Table 10: **Threshold tuning.** Tuned $\tau$ is denoted with $\tau^*$.

| DiffCut | COCO-Object | COCO-Stuff-27 | Cityscapes |
|---|---|---|---|
| $\tau = 0.5$ | 32.0 | 46.1 | 28.4 |
| $\tau^*$ | **38.7** | **48.6** | **29.8** |

As COCO-Object and COCO-Stuff-27 offer different level of object granularity despite corresponding to the same images, a fixed value for $\tau$ can not perform optimally on both benchmarks. Tuning the value of this threshold allows to infer the granularity of objects expected to be uncovered in images. For example, the estimated $\tau^*$ value for COCO-Stuff-27 is 0.35 whereas it is 0.75 on COCO-Object whose annotations requires to detect much finer objects. For Cityscapes the initial fixed $\tau$ value was in the good range to yield optimal performance.

## H    Hungarian Matching

Given a set of predicted masks, our goal is to find the best matching ground-truth labels. For each predicted mask, we compute the Intersection over Union (IoU) with every ground-truth mask and select the one with the highest IoU as its optimal ground-truth match. This results in a pairing where each predicted mask is associated with at most one ground truth mask. In datasets that include a background class, this label implicitly encompasses a variety of concepts related to "things" or "stuff," which vary depending on the dataset. Since our method generates a segment for every detected object in an image, a one-to-one matching between a single predicted cluster and the entire background does not accurately represent the model's true categorization capabilities. Therefore, in such cases, we use a many-to-one matching approach by associating the clusters that primarily overlap with the background to its ID.

# I  Image Noising

Before passing the image to the diffusion UNet, a predefined amount of gaussian noise, controlled by a parameter called the timestep, is added to it. At timestep $t = 0$ the input image corresponds to the original image without added noise while $t = 1000$ corresponds to an image transformed into pure gaussian noise. In Fig. 7, we show the segmentation performance on the validation split of Pascal VOC for timesteps values ranging from 0 to 500. We can observe that a small amount of noise, around 50, gives the best mIoU score, indicating that the best semantic features are obtained with a slightly noisy input image. We note that despite a significant drop in the mIoU score for $t = 500$, DiffCut still reaches state-of-the-art segmentation performance on Pascal VOC benchmark, demonstrating a high robustness of the method with respect to the noising ratio.

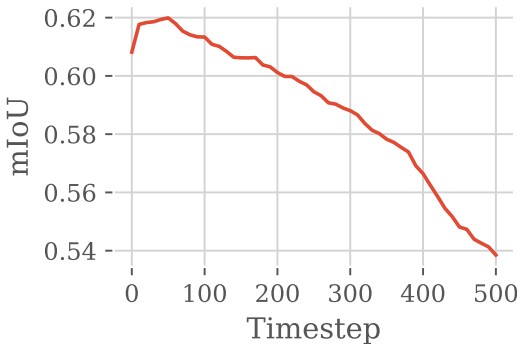

Figure 7: **mIoU according to the noising timestep on Pascal VOC.**

# J  Hyperparameter Sensitivity

Fig. 8 presents an additional evaluation of how the hyperparameters $\alpha$ and $\tau$ influence the segmentation performance on Pascal VOC. Similar to the observations in Fig. 5, we notice a shift in the optimal threshold across the various curves corresponding to different $\alpha$ values. Besides, DiffCut shows increased robustness across a wide range of $\tau$ values, achieving mIoU scores exceeding 60.0 when $\alpha = 10$. In contrast, for $\alpha = 1$, the mIoU only exceeds 60.0 for $\tau$ between 0.91 and 0.94.

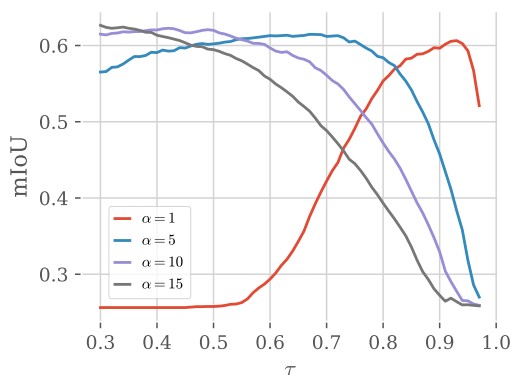

Figure 8: **Robustness of DiffCut on Pascal VOC.**

# K  Runtime Comparison

Tab. 11 presents a runtime comparison between DiffCut, MaskCut, and DiffSeg, which are the main baselines for graph-based image clustering and diffusion-based zero-shot segmentation, respectively.

Table 11: **Runtime Comparison.**

|  | MaskCut ($k = 5$) | DiffCut | DiffSeg - SD1.4 | DiffSeg - SSD-1B |
|---|---|---|---|---|
| images / sec | 0.84 | 1.11 | 2.75 | 1.25 |

## L  Visualization of the effect of $\tau$

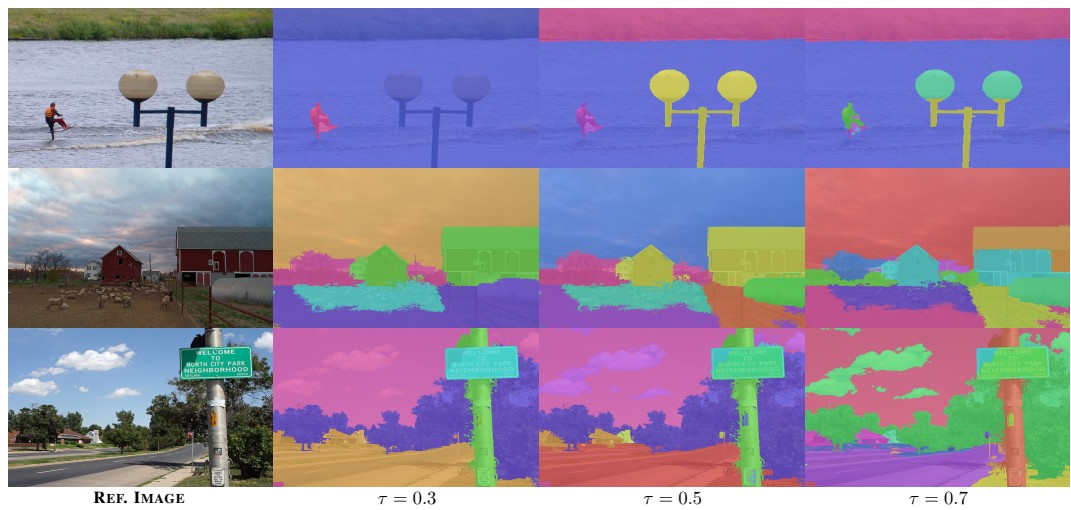

Figure 9: **Effect of $\tau$.** As $\tau$ increases, DiffCut uncover finer-grained objects.

## M  Semantic Coherence in Vision Encoders

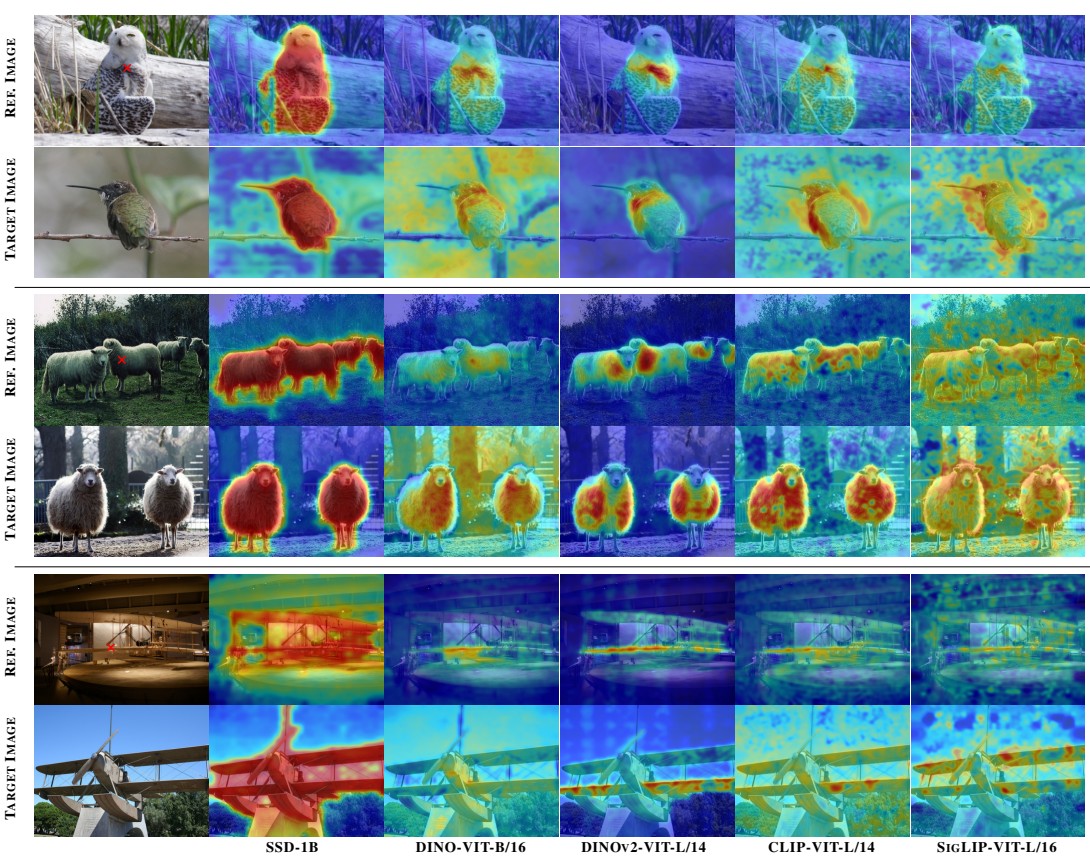

Figure 10: **Qualitative results on the semantic coherence of vision encoders.** We select a patch (red marker) associated to the dog in **REF. IMAGE**. Top row shows the cosine similarity heatmap between the selected patch and all patches produced by vision encoders for **REF. IMAGE**. Bottom row shows the heatmap between the selected patch in **REF. IMAGE** and all patches produced by vision encoders for **TARGET. IMAGE**.

# N   Additional Visualization

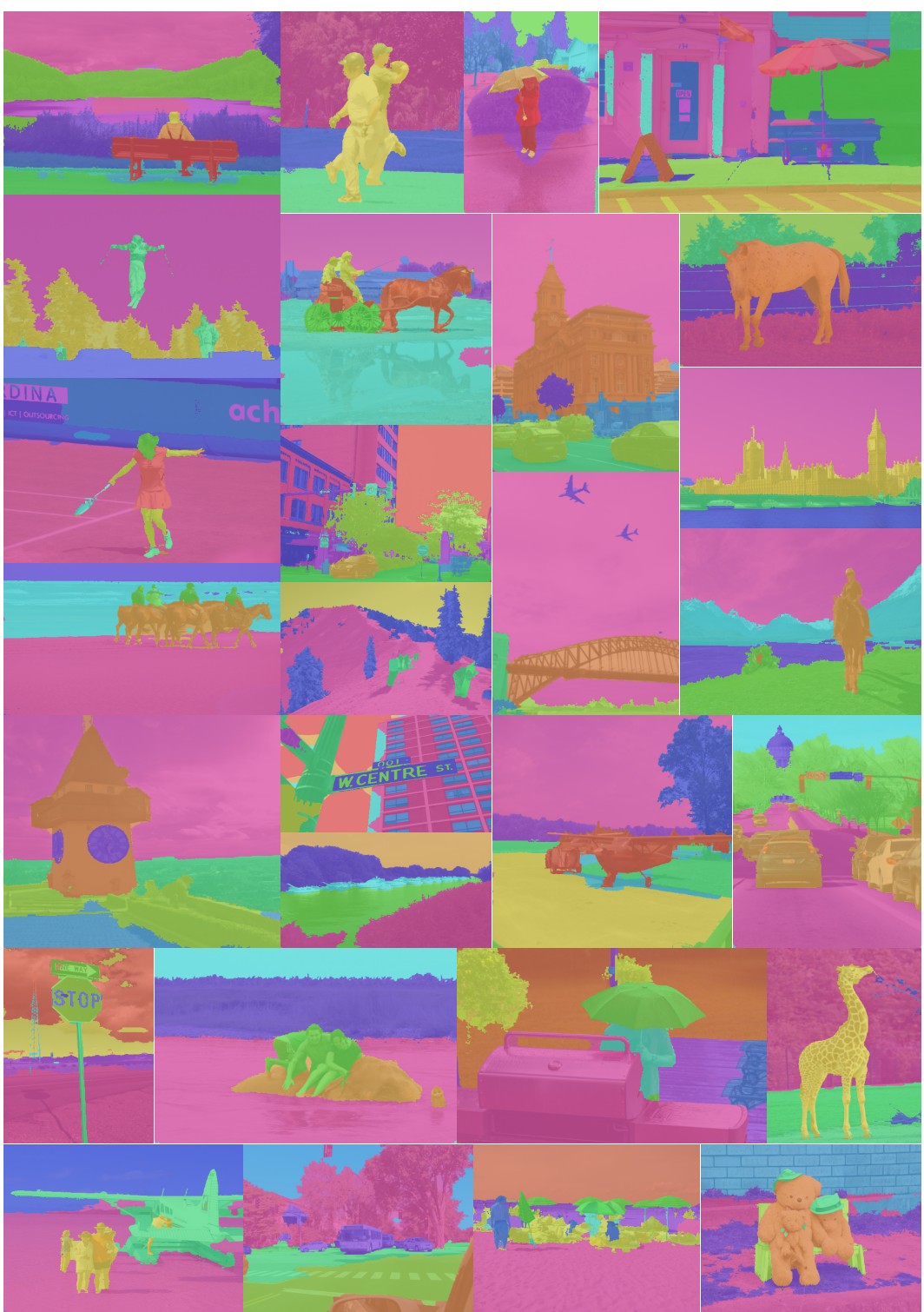

Figure 11: Examples of our produced segmentation maps on COCO dataset.

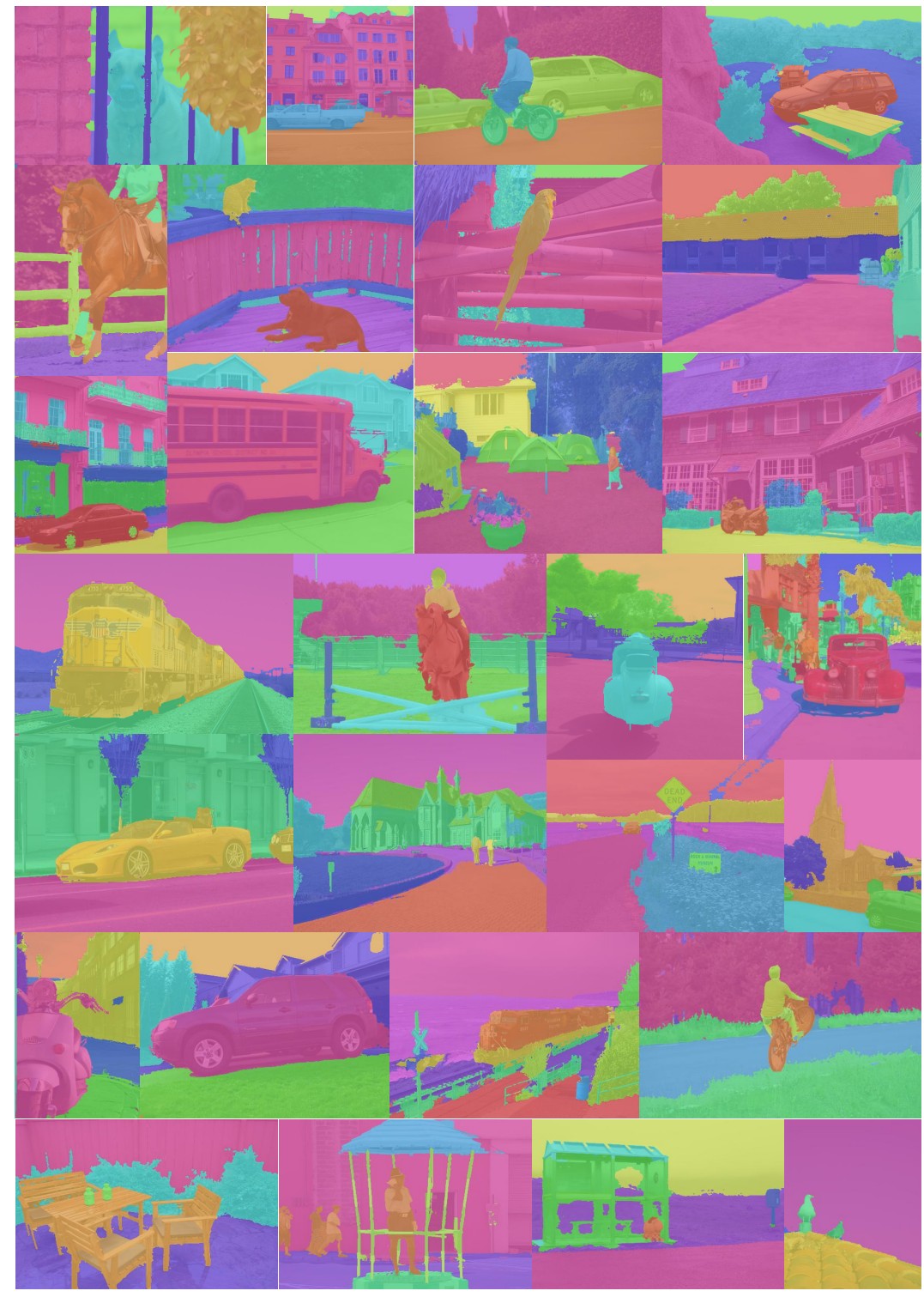

Figure 12: Examples of our produced segmentation maps on Pascal Context dataset.

# O   Datasets Licenses

**Pascal VOC:**   http://host.robots.ox.ac.uk/pascal/VOC/

**Pascal Context:**   https://www.cs.stanford.edu/ roozbeh/pascal-context/

**COCO:**   https://cocodataset.org/#home
License: Creative Commons Attribution 4.0 License

**Cityscapes:**   https://www.cityscapes-dataset.com/
License: This dataset is made freely available to academic and non-academic entities for non-commercial purposes such as academic research, teaching, scientific publications, or personal experimentation.

**ADE20K:**   https://groups.csail.mit.edu/vision/datasets/ADE20K/
License: Creative Commons BSD-3 License

