# OpenReview forum: "DiffCut: Catalyzing Zero-Shot Semantic Segmentation with Diffusion Features and Recursive Normalized Cut"
_NeurIPS.cc/2024/Conference — NeurIPS 2024 poster_

### Official Review · Reviewer_2hwn · 2024-07-09

**Soundness:** 3
**Presentation:** 2
**Contribution:** 3
**Rating:** 6
**Confidence:** 4

**Summary:**

This paper proposes a model for zero-shot semantic image segmentation based on clustering
feeatures from an off-the-shelf text-to-image diffusion model. Features are extracted
from the U-Net used in the diffusion model. The features are then clustered using
a recursive normalized cut algorithm. The clustering is scaled up to the original
image resolution by semantic guidance from the extracted features. The proposed model
consistently outperforms previous methods on standard benchmarks for image segmentation.

**Strengths:**

- The paper makes a very targeted contribution to the field. An off-the-shelf feature
  extractor is combined with an established clustering algorithm and a novel upscaling
  method. The method is evaluated on standard benchmarks which allows for a clear
  comparison to prior work.
- The model is compared to a range of state-of-the-art methods and outperforms them
  consistently, in most cases by a substantial margin.
- Several ablation studies are conducted to show the impact of the contributions in
  comparison to other diffusion model based segmentation methods.

**Weaknesses:**

- The main contribution explitly states the improvements over the TokenCut and MaskCut
  models, but does not compare to these models in the experiments.
- In my view, the comparison of the respective strengths and weaknesses of different
  base models is not detailed enough. The experiment on semantic coherence targets this
  direction. However, beyond this analysis it would have been interesting to directly
  compare the segmentation results after clustering that are obtained from the different
  base models, both in terms of quantiative metrics and example cases and further
  analyses that show the differences.

**Questions:**

- The paper should state more clearly that it target *semantic segmention* to avoid
  confusion with other types of image segmentation. In my view this would have a
  major impact on the clarity of the paper.
- How large is the computational improvement over previous methods? The differences are
  mentioned in terms of parameter count, but a comparison in terms of runtime would be
  interesting as well.
- In comparison to standard semantic segmentation, the method proposed here predicts
  unlabelled classes. Are the features consistent enough to allow for an easy (e.g.
  linear) mapping to the labelled classes across images?

**Limitations:**

I don't fully agree with the statement that the method does not have potential societal
impacts. For example, as briefly mentioned by the authors elsewhere, resusing pre-trained
models in a zero-shot fashion can save computational and labeling resources. In terms
of energy consumption and human labour, this line of works might have a societal impact.

One limitation in my view stems from the fact that the diffusion model used as a basis
probably does not work well on all types of images. While unsupervised, the approach
followed in the paper is dependent on the capabilities of the foundation model. This is
not mentioned in the paper.

---

> ### Author Rebuttal · Authors · 2024-08-06
>
> We thank the reviewer for the helpful comments and appreciate the positive feedback.
>
> ####  **Weaknesses**
> #### **1.  The main contribution explicitly states the improvements over the TokenCut and MaskCut models, but does not compare to these models in the experiments.**
> Unlike TokenCut or MaskCut, our method can provide dense segmentation maps and adapt the number of detected segments based on the visual content of an image. MaskCut and TokenCut cannot do this because they can only detect a fixed number of segments. (l.101-104). Therefore, these methods are not well-suited for an image segmentation task. Technically, MaskCut uses an iterative graph partitioning method and masks the graph nodes associated with detected objects. As a result, each segment is treated as a single object that cannot be refined once detected, which in practice severely limits its ability to identify a large number of objects. To support this claim and following the reviewer’s suggestion, we provide below a comparison between the performance of MaskCut and DiffCut. Overall, the improvement of DiffCut highlights our two main contributions: the quality of our visual features for semantic segmentation, and the capacity of the recursive Ncut algorithm to adjust the number of segments to the visual content of each image. We would be glad to include these results in the final paper.
>
> |  Model         | VOC20 | Context | COCO-Object | COCO-Stuff-27 | Cityscapes | ADE20K |
> |----------------|-------|---------|-------------|------------|------------|--------|
> | DiffCut (ours) | **62.0**  | **54.1**    | **32.0**        | **46.1**       | **28.4**       | **42.4**   |
> | MaskCut $(k=3)$  | 53.7  | 42.3    | 30.9        | 41.8       | 18.0       | 33.7   |
> | MaskCut $(k=5)$  | 53.8  | 43.4    | 30.1        | 41.7       | 18.7       | 35.4   |
> | MaskCut $(k=20)$ | 53.8  | 43.5    | 30.0        | 41.5       | 18.0       | 35.6   |
>
> #### **2.  In my view, the comparison of the respective strengths and weaknesses of different base models is not detailed enough. [...]**
>
> In addition to the experiments conducted in Section 4.2 (Fig. 3 and 4), we evaluated in Supplementary D the potential of different base models for zero-shot segmentation using a simple KMeans clustering on their features for various datasets. This comparison reveals that among the different vision encoders assessed, the diffusion model shows the greatest potential for zero-shot segmentation, as the quality of clustering on its features consistently outperforms other backbones. Given this, we believe that comparing segmentation results after clustering obtained from different base models would yield similar observations.
>
> #### **Questions**
> #### **1.  The paper should state more clearly that it targets semantic segmentation to avoid confusion with other types of image segmentation. In my view this would have a major impact on the clarity of the paper.**
>
> We thank the reviewer for this helpful comment. We will make sure to clearly state and emphasize that this paper targets semantic segmentation to avoid confusion with other types of segmentation.
>
> #### **2.  How large is the computational improvement over previous methods? The differences are mentioned in terms of parameter count, but a comparison in terms of runtime would be interesting as well.**
>
> The primary bottleneck in graph-cut methods is the eigensystem solving which requires $O(n^3)$ operations. In MaskCut, the graph size remains constant, whereas in DiffCut, it has its maximum size only during the first iteration of the recursive NCut. For each subsequent partition, only the relevant nodes from the original graph are selected, reducing the eigensystem's cost. The concept assignment consists in multiplying the matrix of concepts with the feature map and retrieving the indices of the concepts with the highest similarity.
>
> Following your remark, we provide in the general response a table comparing the runtime execution of MaskCut, DiffCut and DiffSeg. We will be glad to include these results in the final manuscript.
>
> #### **3.  In comparison to standard semantic segmentation, the method proposed here predicts unlabelled classes. Are the features consistent enough to allow for an easy (e.g. linear) mapping to the labelled classes across images?**
>
> We have not explored this path yet. However, based on the results in Fig. 3 and 4, we believe that such a mapping would work well, as features from different images belonging to the same semantic class show high similarity. Consequently, we estimate that mapping these features to a class label should be consistent.
>
> ####  **Limitations**
> #### **1.  I don't fully agree with the statement that the method does not have potential societal impacts. [...]**
> Thank you for pointing this out. You are correct that our method which reuses pre-trained models in a zero-shot fashion can indeed save computational ressources as well as reduce energy consumption and human labor, contributing to more sustainable AI practices. We appreciate your remark and will ensure to highlight this in our paper.
>
> ##### **2.  One limitation in my view stems from the fact that the diffusion model used as a basis probably does not work well on all types of images. [...]**
> Since the diffusion backbone is not specifically trained to generate images in specialized domains like biomedical imaging for example, the method may struggle to transfer to out-of-distribution images. However, this issue could potentially be mitigated by fine-tuning the pre-trained diffusion model on the target data domain. As this limitation was not addressed in the main paper, we will discuss it in a Limitations section added to the Supplementary.

---

> > ### Comment · Reviewer_2hwn · 2024-08-09
> >
> > I thank the authors for their detailed response. The additional comparisons reported
> > here and in the responses to the other reviewers as well as the clarifications are very
> > helpful in my view. I agree with with reviewer jdn9 that providing steps towards
> > instance or panoptic segmentation would have increased the impact of the paper.
> > Nevertheless, I think the authors demonstrated the potential of diffusion model features
> > in this setting and overall addressed my concerns, so I am happy to increase my score.

---

### Official Review · Reviewer_oADK · 2024-07-12

**Soundness:** 3
**Presentation:** 3
**Contribution:** 2
**Rating:** 6
**Confidence:** 4

**Summary:**

This work proposes a new strategy for the task of unsupervised zero-shot segmentation using diffusion model features. The semantic maps are extracted from the U-net features of a diffusion model by applying a recursive algorithm that allows various levels of granularities of the segmentation maps. The proposed method beats competing methods on unsupervised and open-vocabular semantic segmentation on six datasets.

**Strengths:**

In general, the work addresses a problem of unsupervised zero-shot segmentation that has not been addressed a lot by the community, proposing new ways to extract knowledge from the rich U-Net features of a diffusion model. While the method is not overly complex and mainly builds on previously published works, it nicely connects more conventional non-DL research from the past with the opportunities arising in foundational models, and it adds a few more technical contributions that are demonstrated to improve performance. In general, the method seems to clearly outperform previous methods on various benchmarks, including the open-vocabulary setting.
The paper is clearly written, and the structure allows to easily follow the work’s main components and contributions.

**Weaknesses:**

My main concern is that the experimental comparison seems rather unfair. While the proposed method uses powerful distilled SD-XL model the baselines only use SD1.4, which might explain most of the performance improvements. In my opinion, the ablation with respect to SD1.4 is not comprehensive enough: 1) The proposed method is not applied with SD1.4, which I think would be mandatory. 2) The DiffSeg results in Tab. 2 seem to be worse than in Tab. 1. I would appreciate a clarification of the effect of using SD-XL features. The proposed method should also be evaluated using the less powerful SD1.4 features to allow a direct comparison with the original DiffSeg method. Otherwise, the comparisons cannot be really fair, in my opinion.

Tab. 4: I assume, DiffSeg would again underperform DiffCut. However, having DiffSeg in this table would be a more complete comparison, not only relaying on the Hungarian algorithm-based evaluation strategy in Tab. 1.

When considering Fig. 9, the masks do get more fine-granular for higher tau. However, the masks do not always align with the object boundaries (e.g. the surfer, the roofs, or the sign). I am wondering whether this is because of using the low-dimensional resolution and only bi-linearly upsampling?

While the previous method DiffSeg also pursued this strategy, I am unsure if the application of the Hungarian matching algorithm to assign predicted masks to a ground truth mask is fair when comparing with methods that do not apply this strategy for the evaluation. It appears not directly comparable. I would appreciate a discussion of this fact and an argumentation why the presented evaluation strategy is reasonable and comparable to the other methods (that do not apply the Hungarian algorithm).

I like the additional baseline AutoSC, which is also seems to be very effective. While the recursive and adaptive formulation of the baseline seem so to be reasonable, the gap to DiffCut is, however, not very clear. The results are almost the same. I would appreciate a more complete ablation of this. The set of chosen exponents seems to be arbitrary (l.302). Did you study whether AutoSC outperforms DiffCut if more possible alpha values were chosen? I would appreciate such an analysis and discussion of it in the rebuttal. Also, I am wondering – which model is faster, AutoSC or DiffCut?

**Questions:**

ReCO is first listed training-free, in the open-vocabulary setting with extra-training, but this is not explained in the text?

Regarding terminology: The section 4.4) model analysis is nicely to read. I would maybe consider replacing the term robustness with hyperparameter sensitivity (also in G). But I have no strong opinion on this.

Tab. 3: Did you also study the results if only 64x64 resolution were used?

l.339: Could you please discuss this strategy more comprehensively in the final manuscript (or refer to a related work if existing)? It is not very clear.

**Limitations:**

The method seems to be profound. However, some important evaluations are missing, as explained above.

---

> ### Author Rebuttal · Authors · 2024-08-06
>
> We thank the reviewer for the helpful comments and appreciate the positive feedback.
>
> #### **Weaknesses**
> #### **1.  [...] The proposed method is not applied with SD1.4, which I think would be mandatory. [...] to allow a direct comparison with the original DiffSeg method. [...].**
> As mentioned in the general response, we emphasize that the ablation study carried out in Tab. 2 uses the same diffusion backbone (SSD-1B) for DiffCut and Diffseg, making the comparison between both approaches fair. Therefore, the large gain of DiffCut over DiffSeg in Tab. 2 directly validates our two main contributions: the choice of the diffusion features and the clustering algorithm. Following the reviewer’s suggestion, we present in the general response a quantitative comparison between DiffCut with SD1.4 and the original DiffSeg method.
>
> #### **2.  The DiffSeg results in Tab. 2 seem to be worse than in Tab. 1. I would appreciate a clarification of the effect of using SD-XL features.**
> The discrepancy between the results of DiffSeg in Tab. 1 and Tab. 2 can be explained by the different backbones used as well as the absence of the refinement module in Tab. 2 as mentioned at lines 293-294. As DiffSeg relies on an attention aggregation process and iterative attention merging, its performance depends on the architecture of the backbone, especially the number and placement of attention modules which significantly differ between SSD-1B and SD1.4.
>
> #### **3.  [...] I assume, DiffSeg would again underperform DiffCut. However, having DiffSeg in this table would be a more complete comparison [...].**
> We agree that having DiffSeg in this table would be a more complete comparison with this particular method. Nevertheless, having demonstrated in previous sections the superiority of DiffCut for generating high-quality masks, the aim here is rather to demonstrate that a straightforward extension to an open-vocabulary setting is competitive with existing approaches targeting this task.
>
> #### **4.  When considering Fig. 9, the masks [...] do not always align with the object boundaries [...]**
> The masks not always aligning perfectly with object boundaries can indeed be attributed to using low-dimensional resolution and relying solely on bi-linear upsampling. The low-dimensional resolution does not capture the fine boundaries of objects, and bi-linear upsampling does not refine the edges to match the true boundaries. Exploring more sophisticated feature upsampling techniques could mitigate this misalignment.
>
> #### **5.  [...] I am unsure if the application of the Hungarian matching [...] is fair when comparing with methods that do not apply this strategy [...]**
> All baselines but ACSeg, ReCO and MaskCLIP are evaluated with Hungarian matching, which is the standard evaluation protocol for methods that provide unlabeled clusters. While ACSeg also evaluates with Hungarian matching, we reported scores with labeled clusters, which are significantly higher (47.1 vs. 53.9 on VOC and 16.4 vs. 28.1 on Stuff-27). Finally, while ReCO and MaskCLIP may not seem directly comparable, we believe that assigning labels to predicted masks can offer more advantageous performance. Hungarian matching performs a one-to-one assignment between each ground-truth label and the single predicted cluster with the greatest overlap, meaning some predicted clusters may not be matched with any ground-truth label. While predicting a class may seem more challenging, it allows assigning a label to every predicted cluster, giving each a chance to be correctly classified, which can ultimately lead to better performance.
>
> #### **6.  I like the additional baseline AutoSC [...] which model is faster, AutoSC or DiffCut?**
> AutoSC is a variant of DiffCut where we replace the recursive clustering by the proposed automated spectral clustering that automatically estimates the number of clusters. Despite its simplicity, it is highly effective and demonstrates competitive performance. Differences mainly appear in datasets containing small objects (e.g., COCO-Object, Cityscapes, ADE20K). DiffCut's recursive partitioning offers flexibility in detecting small objects by allowing large segments to be further divided into smaller ones. In contrast, AutoSC is effective at uncovering the most salient clusters but may overlook smaller objects.
>
> For the set of exponents, we chose the same set we explored for DiffCut. We also examined AutoSC's behavior with higher alpha values, but this did not lead to performance improvements as the index of the largest relative eigen-gap generally corresponds to what is found for alpha in $\\{1, 5, 10, 15\\}$. Lastly, since AutoSC does not require a recursive partitioning of the graph, it benefits from lower runtime execution.
>
> #### **Questions**
> #### **1. ReCO is first listed training-free [...]?**
> We thank the reviewer for pointing this out. This is an oversight on our part; ReCO and MaskCLIP should be listed as training-free methods since the reported scores do not involve any training. We will update the table accordingly.
>
> #### **2. [...] I would maybe consider replacing the term robustness with hyperparameter sensitivity (also in G). [...]**
> We appreciate the reviewer's suggestion and agree that this terminology is more appropriate for these sections. We will update the paper accordingly.
>
> #### **3. [...] Did you also study the results if only 64x64 resolution were used?**
> We have not analyzed the results using only  $64 \times 64$ resolution. Since the image semantics are concentrated in the layers near the UNet’s bottleneck, with the outer layers focusing more on details, we would expect the performance to be less favorable with only $64 \times 64$ resolution.
>
> #### **4. l.339: Could you please discuss this strategy more comprehensively in the final manuscript [...].**
> We thank the reviewer for this valuable feedback. We will discuss this strategy more comprehensively in the final manuscript with complementary details in Supplementary.

---

> > ### Comment · Reviewer_oADK · 2024-08-12
> >
> > Thank you for the detailed response and additional experiments and clarifications. Considering the other reviews and the rebuttal of the authors, I am happy to increase my score 5->6. This paper presents solid work with a fair contribution that should be presented to the community.

---

### Official Review · Reviewer_qiwu · 2024-07-13

**Soundness:** 3
**Presentation:** 2
**Contribution:** 3
**Rating:** 6
**Confidence:** 3

**Summary:**

This paper addresses harnessing semantic localization information in pretrained diffusion UNet models. With the proposed recursive normalized cut on the final self-attention features of the diffusion UNet encoder, this paper achieves better performance on unsupervised zero-shot segmentation comparing to previous methods utilizing other fundation vision encoders, with regulation to the granularity of details. Experiments on several benchmarks and ablation studies show the efficacy of the proposed method.

**Strengths:**

This paper addresses an interesting and important problem of analyzing semantic information in pretrained generative diffusion models comparing to other fundation models. Conducting semantic clusterting on the latents of UNet encoder to reduce computation cost and then applying concept assignment while upsampling seems reasonable. The experiments are overall comprehensive and informative. The emperical results significantly outperform pervious baseline method.

**Weaknesses:**

My main concerns consists of several aspects:
1. The writing seems a bit too vague, maybe with more visual illustraition of the observations? For example, can you demonstrate the "patch-level alignment" with visual examples? It's hard to imagine when reading this in Line 168.
2. Are there any visual verification and description for what kind of concepts you get in Section 3.3?
3. The experiments with the proposed DiffCut seems to mainly excute with SSD-1B. What about using other SD models? Regarding computation cost and performance benefit.

**Questions:**

1. In Line 223, what do you mean by "perform a many-to-one matching"?
2. Are the hyperparameters the same for DiffCut on different datasets? Since you seem to use the same hyperparameters for DiffSeg?

**Limitations:**

This author did not discuss the limitations in the paper, which potentially could include:
1. How is the computation efficiency of the proposed method comparing to other methods? Considering the recursive partitioning and concept assigning process.
2. The semantic clustering knowledge in pretrained diffusion models might not be able to transfer to data domains like biomedical images.
3. If LD, AX, or UA is given, would the proposed method still outperform the corresponding SOTAs? Or can it further improve its performance?

---

> ### Author Rebuttal · Authors · 2024-08-06
>
> We thank the reviewer for the helpful comments and appreciate the positive feedback.
>
> #### **Weaknesses**
> ####  **1.  Writing is a bit too vague [...]. For example, can you demonstrate the "patch-level alignment" with visual examples?**
> For vision tasks, we expect the vision encoder to be semantically coherent i.e to produce similar patch representations for regions belonging to the same semantic concept. We refer to it as the “patch-level alignment”. We show in Fig 4. and Fig 10.  some qualitative results that illustrate the inner patch-level alignment for different vision encoders. We visualize the patch level alignment by encoding an image and computing the cosine similarity between a reference patch and all other patches. For SSD-1B, in Fig 4. the dog is uniformly and strongly highlighted, indicating a strong patch-level alignment. Following the reviewer’s remark, we will refer to Fig.4 at l.168 for better clarity on this point.
> #### **2.  Are there any visual verification and description for what kind of concepts you get?**
> In section 3.3, each concept is obtained via a reduction of the spatial dimension of the corresponding segment (Masked SMM l.195) . This way, a single concept has the form of a feature vector that globally describes its corresponding segment. Providing a visual description of a single concept is not straightforward and would imply a form of “decoding” of this concept.
>
> #### **3.  The experiments [...] execute with SSD-1B. What about using other SD models? [...]**
> Following the reviewer’s recommendation, we conduct a new evaluation to demonstrate that DiffCut provides competitive performance with other smaller diffusion backbones than SSD-1B. Specifically, we use SD1.4 and SSD-vega (another distilled version of SDXL). For SD1.4 the UNet encoder has 260M parameters (~ 30% of the UNet), for SSD-vega, the UNet encoder has 240M parameters (~32% of the UNet).
>
> |  Model   | VOC20 | Context | COCO-Object | COCO-Stuff | Cityscapes | ADE20K |
> |----------|-------|---------|-------------|------------|------------|--------|
> | SD1.4    | 57.5  | 52.8    | 30.0        | 45.2       | 24.5       | 36.7   |
> | SSD-Vega | 62.2  | 56.4    | 34.9        | 49.5       | 30.1       | 45.7   |
> | SSD-1B   | 65.2  | 56.5    | 34.1        | 49.1       | 30.6       | 44.3   |
> | DiffSeg  | 49.8  | 48.8    | 23.2        | 44.2       | 16.8       | 37.7   |
>
> The results obtained with these two backbones are consistent with those achieved using SSD-1B. Although there is a slight performance drop with the SD1.4 encoder backbone compared to SSD-1B, the method still significantly outperforms DiffSeg. Remarkably, DiffCut with SSD-vega UNet encoder achieves performance comparable to SSD-1B, despite being only half its size. We will be glad to add these results in the final paper.
>
> #### **Questions**
> #### **1. What do you mean by "perform a many-to-one matching"?**
> In datasets that include a background class, this label implicitly encompasses a variety of concepts related to "things" or "stuff," which vary depending on the dataset. Since our method generates a segment for every detected object in an image, a one-to-one matching between a single cluster ID and the entire background's ID does not accurately represent the model's true categorization capabilities. Therefore, in such cases, we use a many-to-one matching approach by associating the clusters that primarily overlap with the background to its ID. We will be glad to include this clarification in the final paper.
>
> #### **2.  Are the hyperparameters the same for DiffCut on different datasets?**
> As highlighted in the general response and noted in the Implementation Details section (l.230), we used a fixed set of hyperparameters, $\tau$ and $\alpha$, across all datasets for our evaluations. This approach ensures a fair comparison against the baselines and is particularly important for our evaluation of DiffSeg, where we also maintain a single hyperparameter value across different datasets.
>
> #### **Limitations**
> #### **1.  How is the computation efficiency of the proposed method comparing to other methods? [...]**
> The primary bottleneck in graph-cut methods is the eigensystem solving which requires $O(n^3)$ operations with $n$ being the number of nodes in the graph. In MaskCut, the graph size remains constant, whereas in DiffCut, it has its maximum size only during the first iteration of the recursive NCut. For each subsequent partition, only the relevant nodes from the original graph are selected, reducing the eigensystem's cost. The concept assignment essentially consists in multiplying the matrix of concepts with the feature map.
>
> Following your remark, we provide in the general response a table comparing the runtime execution of MaskCut, DiffCut and DiffSeg. We will be glad to include these results in the final manuscript.
>
> #### **2.  The semantic clustering knowledge in pretrained diffusion models might not be able to transfer to data domains like biomedical images.**
> Since the diffusion backbone is not specifically trained to generate images in specialized domains like biomedical imaging, the method may struggle to transfer to out-of-distribution images. However, this issue could potentially be mitigated by fine-tuning the pre-trained diffusion models on the target data domain. As this limitation was not addressed in the main paper, we will discuss it in a Limitations section added to the Supplementary.
>
> #### **3.  If LD, AX, or UA is given, would the proposed method still outperform the corresponding SOTAs? [...]**
> Since our method is entirely training-free and does not rely on LD, AX, or UA, we believe that incorporating any additional information could further enhance its performance. Specifically, given the superior quality of our masks compared to MaskCut, we believe that integrating our DiffCut method to generate high-quality pseudo-masks within an unsupervised training scheme, would lead to even better performance.

---

> > ### Comment · Reviewer_qiwu · 2024-08-14
> >
> > I thank the author for the detailed response. After reading the reviews and the rebuttal, I agree this is a good work and would like to increase my rating.

---

### Official Review · Reviewer_jdn9 · 2024-07-13

**Soundness:** 2
**Presentation:** 3
**Contribution:** 2
**Rating:** 5
**Confidence:** 5

**Summary:**

This paper introduces an innovative unsupervised, zero-shot image segmentation method called DiffCut. This method leverages the encoder features of a pre-trained diffusion model within a recursive graph partitioning algorithm to create finely detailed segmentation maps without requiring labels from downstream segmentation datasets. DiffCut exploits features from the last self-attention block of a diffusion UNet encoder to perform image segmentation. This approach does not rely on paired image and text data, making it suitable for unsupervised and zero-shot learning tasks. The core algorithmic innovation in DiffCut is the use of a recursive Normalized Cut (NCut) that allows the model to regulate the granularity of detected objects and consequently adapt the number of segments to the visual content of each image. Compared to existing methods like DiffSeg and other graph-based object localization techniques, DiffCut significantly outperforms in terms of the quality of segmentation maps and alignment with semantic visual concepts. The effectiveness of DiffCut was validated across multiple standard benchmarks with a focus on mIoU scores, where it consistently outperformed the state-of-the-art unsupervised semantic segmentation methods.

**Strengths:**

1. The paper is clearly written and logically structured, making it easy to understand.

2. The authors conducted thorough experiments to demonstrate the effectiveness of DiffCut, which achieves state-of-the-art performance in many downstream tasks.

3. In contrast to earlier works that employ graph-based clustering methods for unsupervised segmentation, such as TokenCut and MaskCut, this method introduces a soft thresholding technique for constructing the affinity matrix. This approach effectively maintains high affinity between highly similar patches while reducing the weights between dissimilar patches to near zero—an improvement that addresses a limitation often neglected in previous studies.

**Weaknesses:**

1. **[Unsupervised Instance or Panoptic Segmentation Performance]** While the paper positions itself as effective in segmenting more objects within an image, it primarily focuses on semantic segmentation, potentially overlooking the model's capability for instance discrimination. I am interested in how the model performs in unsupervised instance or panoptic segmentation tasks, as these require differentiating individual objects or integrating both semantic and instance segmentation.

2. **[Technical Contribution and Comparison with MaskCut]** MaskCut is actually capable of segmenting multiple objects per image, with the ability to set a large number of masks per image. This raises questions about the novelty and technical contributions of this paper, particularly in the absence of quantitative comparisons with established methods like TokenCut [30] or MaskCut [32].

3. **[Lack of Comparisons with Previous Works]** The paper does not compare its results with some previous state-of-the-art works in unsupervised panoptic and semantic segmentation, such as U2Seg [N1] and EAGLE [N2]. Notably, [N2] also explores the use of prototypes derived from eigenvectors of feature maps.

4. **[Adequacy of Low-Resolution Features]** The use of low-resolution feature maps to produce concept-embeddings may result in missing small-scale objects in the images, which could lead to the exclusion of these small objects during the "High-Resolution Concept Assignment" phase. This could limit the model’s effectiveness in capturing medium or small-sized objects, which are crucial for detailed image segmentation.

5. **[Recursive Normalized Cuts or Multi-class Spectral Clustering]** I am curious if the authors have explored to use spectral clustering for multi-entity/object segmentation. The granularity of the segmentation masks can be controlled by using a different cluster numbers.


[N1] Niu, Dantong, Xudong Wang, Xinyang Han, Long Lian, Roei Herzig, and Trevor Darrell. "Unsupervised universal image segmentation." In Proceedings of the IEEE/CVF Conference on Computer Vision and Pattern Recognition, pp. 22744-22754. 2024.

[N2] Kim, Chanyoung, Woojung Han, Dayun Ju, and Seong Jae Hwang. "EAGLE: Eigen Aggregation Learning for Object-Centric Unsupervised Semantic Segmentation." In Proceedings of the IEEE/CVF Conference on Computer Vision and Pattern Recognition, pp. 3523-3533. 2024.

**Questions:**

Please check the weakness section.

**Limitations:**

The authors adequately addressed the limitations.

---

> ### Author Rebuttal · Authors · 2024-08-06
>
> We thank the reviewer for the helpful comments and appreciate the positive feedback.
>
> #### **Weaknesses**
> #### **1. [Unsupervised Instance or Panoptic Segmentation Performance]**
> We have not yet evaluated DiffCut’s performance in instance or panoptic segmentation. The goal here is to demonstrate that the features of a diffusion UNet surpass those of other vision backbones in precise semantic segmentation. Given that our method can generate high-quality semantic masks under the most restrictive conditions (training-free and independent of LD, AX, and UA), future work could involve relaxing these constraints and incorporating DiffCut as a high-quality pseudo-label generator within an unsupervised training pipeline—such as proposed in U2Seg [1]. This could significantly enhance performance in unsupervised semantic, instance, and panoptic segmentation.
>
> #### **2. [Technical Contribution and Comparison with MaskCut]**
> Unlike TokenCut or MaskCut, our method can provide dense segmentation maps and adapt the number of detected segments based on the visual content of an image. MaskCut and TokenCut cannot do this because they can only detect a fixed number of segments. (l.101-104). Therefore, these methods are not well-suited for an image segmentation task. Technically, MaskCut uses an iterative graph partitioning method and masks the graph nodes associated with detected objects. As a result, each segment is treated as a single object that cannot be refined once detected, which in practice severely limits its ability to identify a large number of objects. To support this claim and following the reviewer’s suggestion, we provide below a comparison between the performance of MaskCut and DiffCut. Overall, the improvement of DiffCut highlights our two main contributions: the quality of our visual features for semantic segmentation, and the capacity of the recursive Ncut algorithm to adjust the number of segments to the visual content of each image. We would be glad to include these results in the final paper.
>
> |  Model         | VOC20 | Context | COCO-Object | COCO-Stuff-27 | Cityscapes | ADE20K |
> |----------------|-------|---------|-------------|------------|------------|--------|
> | DiffCut (ours) | **62.0**  | **54.1**    | **32.0**        | **46.1**       | **28.4**       | **42.4**   |
> | MaskCut $(k=3)$  | 53.7  | 42.3    | 30.9        | 41.8       | 18.0       | 33.7   |
> | MaskCut $(k=5)$  | 53.8  | 43.4    | 30.1        | 41.7       | 18.7       | 35.4   |
> | MaskCut $(k=20)$ | 53.8  | 43.5    | 30.0        | 41.5       | 18.0       | 35.6   |
>
>
> #### **3.  [Lack of Comparisons with Previous Works]**
> We thank the reviewers for these recent references. In comparison with EAGLE [2], we propose a much simpler and effective training-free pipeline that reaches higher results on COCO-Stuff and Cityscapes due to the quality of the extracted features from the diffusion backbone. Their pipeline is based on DINOv1 which we proved in Fig. 3 has worse feature correspondence than our backbone. U2Seg [1] proposes a framework to unify unsupervised semantic, instance and panoptic segmentation. This method only evaluates on COCO-Stuff-27 for unsupervised semantic segmentation. We will be glad to add these baselines to our main Tab. 1.
>
> |         | COCO-Stuff-27 | Cityscapes |
> |---------|:-------------:|:----------:|
> | DiffCut |      **49.1**     |    **30.6**    |
> | DiffSeg |      43.6     |    21.2    |
> |  EAGLE  |      27.2     |    22.1    |
> |  U2SEG  |      30.2     |      -     |
>
> #### **4.  [Adequacy of Low-Resolution Features]**
> We empirically observe in Fig. 9, 11 and 12 that the excellent quality of the features provided by the diffusion model at a reasonably low spatial resolution $(32 \times 32)$ effectively enables the detection of medium to small-sized objects. Increasing the value of the hyperparameter $\tau$ can even further help to detect objects at a finer granularity. This spatial resolution of feature maps does not present a significant drawback for small object localization.
>
>
> #### **5.  [Recursive Normalized Cuts or Multi-class Spectral Clustering]**
> In Tab. 2, we explore the use of spectral clustering for multi-object segmentation by introducing a variant of DiffCut called AutoSC. This variant adapts the number of segments based on the visual content using a heuristic called “relative-eigen-gap” which estimates the number of connected components in a graph. This estimated number of connected components, $k$, is then used to determine the number of clusters in $k$-way spectral clustering. We observe that AutoSC also achieves excellent results, often comparable to those obtained with DiffCut.
>
> We note that the main differences between AutoSC and DiffCut appear in datasets containing small objects (e.g., COCO-Object, Cityscapes, ADE20K). DiffCut’s recursive partitioning offers flexibility in detecting small objects by allowing large segments to be further divided into smaller ones, while the “relative eigen-gap” heuristic in AutoSC is effective at uncovering the most salient clusters, potentially overlooking smaller objects.
>
> [1] Niu, Dantong, Xudong Wang, Xinyang Han, Long Lian, Roei Herzig, and Trevor Darrell. "Unsupervised universal image segmentation." In Proceedings of the IEEE/CVF Conference on Computer Vision and Pattern Recognition, pp. 22744-22754. 2024.
>
> [2] Kim, Chanyoung, Woojung Han, Dayun Ju, and Seong Jae Hwang. "EAGLE: Eigen Aggregation Learning for Object-Centric Unsupervised Semantic Segmentation." In Proceedings of the IEEE/CVF Conference on Computer Vision and Pattern Recognition, pp. 3523-3533. 2024.

---

> > ### Comment · Reviewer_jdn9 · 2024-08-12
> >
> > I appreciate the authors for addressing most of my concerns in the rebuttal. I will keep my rating as borderline accept. I encourage the authors to include the additional comparisons, especially the comparisons with MaskCut in the main paper, which can help other researchers to better understand the key distinctions and benefits of the proposed method.

---

### Author Rebuttal · Authors · 2024-08-06

We thank the reviewers for their helpful comments and valuable suggestions. We would like to clarify here some key points raised by the reviewers and we then provide individual responses to each one.

### **Comparison with MaskCut**

Reviewers jdn9 and 2hwn mentioned that although we positioned our method as an improvement over MaskCut, we did not quantitatively compare against it. In the specific answers to R.jdn9 and R.2hwn, we explain why MaskCut was not initially designed for semantic segmentation, in contrast to state-of-the-art methods evaluated in Tab. 1. To further analyze the potential of MaskCut in semantic segmentation and meet R.jdn9 and R.2hwn’s request, we evaluate this method on the semantic segmentation task. Specifically, as MaskCut requires a predefined number of iterations $k$ to detect a fixed number of segments per image,  we evaluate MaskCut with $k$ in $\\{3, 5, 20\\}$. The quantitative comparison between DiffCut and MaskCut is shown below, and we would be glad to include it in the final paper.

|  Model         | VOC20 | Context | COCO-Object | COCO-Stuff-27 | Cityscapes | ADE20K |
|----------------|-------|---------|-------------|------------|------------|--------|
| DiffCut (ours) | **62.0**  | **54.1**    | **32.0**        | **46.1**       | **28.4**       | **42.4**   |
| MaskCut $(k=3)$  | 53.7  | 42.3    | 30.9        | 41.8       | 18.0       | 33.7   |
| MaskCut $(k=5)$  | 53.8  | 43.4    | 30.1        | 41.7       | 18.7       | 35.4   |
| MaskCut $(k=20)$ | 53.8  | 43.5    | 30.0        | 41.5       | 18.0       | 35.6   |

We can see that DiffCut significantly and consistently outperforms MaskCut across all datasets for any chosen $k$. These additional experiments further highlight the contributions of our DiffCut method, i.e., the diffusion features used for segmentation, and the recursive NCut method able to adapt the number of segments to the visual content of each image. In addition, we observe no significant improvements for $k=5$ or $k=20$ in comparison with $k=3$, supporting the claim that MaskCut is inadequate to detect a large number of objects due to its iterative process.


### **Clarifications in Experiments**
#### **1. Comparison to DiffSeg.**
Reviewers qiwu and oADK raised concerns about the fairness in the comparison of our DiffCut method with respect to the DiffSeg baseline. First, R.qiwu asked if a common hyperparameter setting has been used for all datasets with DiffCut, as it is the case for DiffSeg. We highlight that we used a fixed set of hyperparameters for DiffCut across all datasets for our evaluations, ensuring a fair comparison with DiffSeg, as mentioned in lines 232-233.

Additionally, R.oADK suggested that the comparison might be unfair due to the different backbones used by our method and DiffSeg. We emphasize that the ablation study carried out in Tab. 2 uses the same diffusion backbone (SSD-1B) for DiffCut and Diffseg, making the comparison between both approaches fair. Therefore, the large gain of DiffCut over DiffSeg in Table 2 directly validates our two main contributions: the choice of the diffusion features (vs self-attention maps in DiffSeg), and the clustering algorithm (recursive NCut vs iterative attention merging).

To explicitly meet R.oADK’s request, we conducted an additional comparison between DiffCut and DiffSeg, using the SD1.4 backbone as employed in the original DiffSeg method. The results shown below confirm that even with SD1.4, DiffCut significantly and consistently outperforms DiffSeg across all datasets but ADE20K where the difference of only 1pt. We will be glad to add these new results to the final paper to emphasize on the fairness of the conducted evaluation.

|  Model           | VOC20 | Context | COCO-Object | COCO-Stuff-27| Cityscapes | ADE20K |
|------------------|-------|---------|-------------|------------|------------|--------|
| DiffSeg w/ SD1.4 | 49.8  | 48.8    | 23.2        | 44.2       | 16.8       | **37.7**   |
| DiffCut w/ SD1.4 | **57.5**  | **52.8**    | **30.0**        | **45.2**       | **24.5**       | 36.7   |

#### **2. Evaluation Strategy.**
On R.oADK’s concern about the fairness of our evaluation strategy, we highlight that all methods in Tab. 1 are evaluated using Hungarian matching but ACSeg, ReCO and MaskCLIP which attribute labels to predicted masks. For these latter methods, we clarify that the Hungarian matching may leave some predicted clusters unassigned to any ground truth labels whereas assigning text labels offers a chance to each cluster to be correctly classified. For example, STEGO and ACSeg, which also evaluate with label predictions, show significantly higher performance in this setup than with Hungarian matching evaluation. Our evaluation setup is thus reasonable, making the comparison with ReCO and MaskCLIP in Tab. 1 fair.

#### **3. Runtime Comparison**
Reviewers  qiwu and 2hwn have asked for a runtime comparison between methods. Following their requests, we provide in the following table a runtime comparison between DiffCut, MaskCut and DiffSeg which are respectively the two main baselines when it comes to graph-based image clustering and diffusion-based zero-shot segmentation.
|                        | MaskCut $(k=5)$ | DiffCut | DiffSeg - SD1.4 | DiffSeg - SSD-1B |
|------------------------|---------------|---------|-----------------|-------------------|
| Images / sec | 0.84          | 1.11    | 2.75            | 1.25              |
|                        |               |         |                 |                   |

MaskCut with $k=5$ is the slowest method, segmenting 0.84 images per second. Using SSD-1B with images at $1024 \times 1024$ resolution, DiffCut’s speed is slightly lower than DiffSeg’s based on the same architecture. With the SD1.4 backbone, DiffSeg demonstrates superior runtime performance due to the smaller size of the architecture and the input image size. We will be glad to include these results in the final manuscript.

---

### Decision · Program_Chairs · 2024-09-25

**Decision:**

Accept (poster)

**Comment:**

The paper is reviewed by 4 reviewers. All reviewers recommend acceptance. The authors should add to the additional comparisons, clarifications and discussions about limitations to the final copy or supplementary materials.